# The Multifaceted Role and Utility of MicroRNAs in Indolent B-Cell Non-Hodgkin Lymphomas

**DOI:** 10.3390/biomedicines9040333

**Published:** 2021-03-25

**Authors:** Pinelopi I. Artemaki, Petros A. Letsos, Ioanna C. Zoupa, Katerina Katsaraki, Paraskevi Karousi, Sotirios G. Papageorgiou, Vasiliki Pappa, Andreas Scorilas, Christos K. Kontos

**Affiliations:** 1Department of Biochemistry and Molecular Biology, Faculty of Biology, National and Kapodistrian University of Athens, 15701 Athens, Greece; partemaki@biol.uoa.gr (P.I.A.); petr.sletsos@gmail.com (P.A.L.); joanz3999@gmail.com (I.C.Z.); kkatsaraki@biol.uoa.gr (K.K.); pkarousi@biol.uoa.gr (P.K.); ascorilas@biol.uoa.gr (A.S.); 2Second Department of Internal Medicine and Research Unit, University General Hospital “Attikon”, 12462 Athens, Greece; sotirispapageorgiou@hotmail.com (S.G.P.); vaspappa@med.uoa.gr (V.P.)

**Keywords:** miRNAs, prognosis, follicular lymphoma, marginal zone lymphoma, Waldenström’s macroglobulinemia, hairy cell leukemia, primary cutaneous follicle center lymphoma, normal B-cell development, therapeutic target, diagnosis

## Abstract

Normal B-cell development is a tightly regulated complex procedure, the deregulation of which can lead to lymphomagenesis. One common group of blood cancers is the B-cell non-Hodgkin lymphomas (NHLs), which can be categorized according to the proliferation and spread rate of cancer cells into indolent and aggressive ones. The most frequent indolent B-cell NHLs are follicular lymphoma and marginal zone lymphoma. MicroRNAs (miRNAs) are small non-coding RNAs that can greatly influence protein expression. Based on the multiple interactions among miRNAs and their targets, complex networks of gene expression regulation emerge, which normally are essential for proper B-cell development. Multiple miRNAs have been associated with B-cell lymphomas, as the deregulation of these complex networks can lead to such pathological states. The aim of the present review is to summarize the existing information regarding the multifaceted role of miRNAs in indolent B-cell NHLs, affecting the main B-cell subpopulations. We attempt to provide insight into their biological function, the complex miRNA-mRNA interactions, and their biomarker utility in these malignancies. Lastly, we address the limitations that hinder the investigation of the role of miRNAs in these lymphomas and discuss ways that these problems could be overcome in the future.

## 1. Introduction

B-cell non-Hodgkin lymphomas (NHLs) are one of the most common malignancies. They consist of different types of lymphomas, which are characterized by great heterogeneity. Their common feature is the absence of Reed-Sternberg cells, which in contrast are present in Hodgkin Lymphomas. B-cell NHLs are more common than the Hodgkin lymphomas and are classified according to the proliferating and spread rate of cancer cells into indolent or slowly growing and aggressive or quickly growing lymphomas. Recent advances have assisted in the elucidation of the etiology and the molecular background of these lymphomas; however, several questions remain unanswered and thus hinder the in-depth understanding of the pathogenesis of these lymphomas and the adoption of a personalized treatment approach. Two of the most common indolent B-cell NHLs are marginal zone lymphoma (MZL) and follicular lymphoma (FL), which sometimes could transform into more aggressive types, such as diffuse large B-cell lymphoma (DLBCL) [1,2]. Besides these two lymphoma types, Waldenström’s macroglobulinemia (WM), hairy cell leukemia (HCL), and primary cutaneous follicle center lymphoma (PCFCL), which occur more rarely, are considered as indolent B-cell NHLs as well (Figure 1).

FL is the most common indolent B-cell NHL and shows great heterogeneity. Its diagnosis is based on the detection of malignant centrocytes and centroblasts that resemble germinal center B cells within lymphoid follicles [3,4]. MZL is less investigated in comparison with FL, while various treatment options are available. It derives from malignant marginal zone B cells and persistent immune system stimulation triggered by infections or autoimmune diseases, which constitutes a major driving factor of lymphomagenesis. MZL is divided into three main subtypes: Extranodal MZL or mucosa-associated lymphoid tissue (MALT) lymphoma, splenic MZL, and nodal MZL, depending on the site where the malignancy originates [1,2,5].

MicroRNAs (miRNAs) are small non-coding RNAs of approximately 22 nucleotides long that can greatly influence protein expression. RNA polymerase II–mediated transcription gives rise to pri-miRNA, which is subsequently cleaved by the nuclear RNase III DROSHA. The resulting pre-miRNA is exported from the nucleus to the cytoplasm via exportin 5 (EXP5), and then the cytoplasmic RNase III DICER1 cleaves the pre-miRNA near the terminal loop. Ultimately, one of the two strands prevails and interacts with proteins of the Argonaute (AGO) family to form the RNA-induced silencing complex (RISC). Targeting of a specific mRNA molecule through base-pairing between the miRNA and the 3′ untranslated region (3′ UTR) results in translational repression or mRNA degradation and, therefore, diminished protein levels [6].

Considering that a miRNA can target multiple mRNAs, that a single mRNA can be targeted by several miRNAs, and that miRNA transcription can be regulated, complex gene expression networks emerge. As will be thoroughly discussed below, some of these networks have been proved to be essential for proper B-cell development and therefore their deregulation can lead to B-cell lymphomas. Multiple miRNAs have been associated with such pathological states, such as those of the miR-17/92 cluster, miR-155-5p, and miR-150-5p, while their targets involve the transcription factors FOXP1, MYC, and MYB, and the affected signaling pathways include the BCR, NFkB, and PI3K/AKT [7,8]. Besides their regulatory potential in B-cell lymphomas, miRNAs have been proposed as potential biomarkers, due to their relatively high stability in biological samples, including bodily fluids and fixed tissues, and their high specificity and sensitivity. Therefore, they could be used for personalized prognosis, prediction of therapeutic response, and as an additional tool for differential diagnosis [9].

In this review, we attempt to summarize the existing information regarding the multifaceted role of miRNAs in the indolent B-cell NHLs. On the one hand, most of the current studies investigate the expression profiles of miRNAs, an endeavor that could lead to novel biomarkers discovery. These biomarkers could be utilized to predict the transformation to a more aggressive entity of the disease, to assist in the correct differential diagnosis and to choose the optional treatment, as well as to monitor the therapeutic response. On the other hand, we attempt to provide insight into the biological function and the complex miRNA-mRNA interactions, and to further elucidate the molecular mechanisms underlying disease progression. Lastly, we address the limitations that hinder the investigation of the role of miRNAs in these lymphomas and discuss ways that these problems could be overcome in the future.

## 2. miRNAs in Normal B-Cell Development

Normal B-cell development is a tightly regulated complex procedure. Briefly, B cells derive from hematopoietic stem cells (HSCs) in the bone marrow, where the first steps of differentiation occur. V(D)J recombination of immunoglobulin (Ig) heavy (IgH) and light (IgL) chain genes facilitate the differentiation from pro-B cell to pre-B cell and can lead to the formation of an immature B cell that expresses a functional B-cell receptor (BCR) with unique specificity. In case these gene rearrangements are unproductive or the BCR binds strongly to presented self-antigens, the B cell cannot complete the central tolerance checkpoint, and therefore B cells are eliminated. After that, migration of B cells to the spleen takes place. In the spleen, naïve B cells can be activated by foreign-antigen recognition. Provided that they do not become autoreactive through somatic hypermutation, they differentiate into follicular or marginal zone B cells. This differentiation strongly depends on BCR signaling. Next, marginal zone B cells reside in the marginal zone, while follicular B cells enter germinal centers, forming three distinct zones: the dark, light, and mantle zone. Finally, germinal center B cells differentiate into memory or plasma cells [10].

### 2.1. miRNAs in Primary Lymphoid Tissue B-Cell Development

The role of miRNAs in normal B-cell development has been variously described. Most interactions between miRNAs and mRNAs involved in normal B-cell development have been investigated in mouse models; however, the vast majority of them have been predicted and/or validated in human cells as well, as miRNAs are highly conserved among species. The great effect of miRNA function in the development of B cells is prominent, as their absence completely abolishes this process; specifically, in Dicer-deficient mice, the developmental procedure was arrested, while mice lacking Dgcr8 showed elevated early B-cell apoptosis [11,12]. Interestingly, most miRNAs show a stage-specific expression pattern, indicating their stage-specific function [13]; relative examples of miRNAs showing stage-specific expression are those of miR-150-5p, miR-181a-5p, miR-126-3p, and miR-132-3p [14,15,16].

The developmental procedure is mainly dictated by transcription factors. Some of them show a stage-related expression as well, while others are essential in every developmental stage. TCF3 suggests a transcription factor vital for the whole developmental process, while EBF1 and PAX5 are essential for specific steps of it, as they are involved in particular processes, including BCR formation [17,18,19,20]. However, EBF1 deficiency does not lead to the eradication of the development of B cells, as the process is rescued by miR-126-3p, which was shown to stimulate the expression of *RAG1* and *RAG2* recombinases that mediate VDJ recombination [15,21]. This explains the necessity for high miR-126-3p levels in the early steps of the process. RAG1 and RAG2 expression is also regulated by the transcription factor FOXP1 [22]; in this sense, murine *Foxp1* and *Tcf3* suggest miR-191-5p targets, and so does *Egr1*, another transcription factor vital for the maturation of B cells, also targeted by miR-146a-5p [23,24]. miR-191-5p has been characterized as a rheostat for the process, as both its higher and lower levels disrupt B-cell development, due to the subsequent changes in transcription factor levels [24].

The above data delineate a miRNA-transcription factor network, showing a great impact on the developmental procedure. Disruption of this network has been variously witnessed to abolish B-cell development at pro- to pre-B cell differentiation stage. A typical such paradigm is that of miR-132-3p, a miRNA normally expressed in late developmental stages. Under physiological circumstances, miR-132-3p expression is BCR-dependent, thus it is abundant after the pro-B stage, when a functional BCR has been developed [25]. When overexpressed in the early stages, the process stopped at pro- to pre-B-cell transition due to *Sox4* transcription factor deficiency, which regulates Rag1 expression, as it was shown in xenografts. In this context, another miRNA showing stage-specific expression, namely miR-150-5p, blocks the developmental process at the same point when expressed prematurely, through *MYB* transcription factor inhibition [14,26]. This transcription factor participates in the proliferation and differentiation of hematopoietic progenitor cells.

Moreover, miR-24-3p, a member of the miR-23a cluster that promotes HSC differentiation towards common myeloid progenitors rather than lymphoid progenitors [27], functions as a *MYC* inhibitor, leading to inhibition of pro- to pre- B-cell transition [28]. MYC transcription factor is considered as a key molecule for B-cell development, as it regulates the expression of the miR-17/92 cluster; members of this cluster target *BCL2L11*, which encodes a pro-apoptotic protein, as well as *PTEN*, a key molecule for the PI3K pathway with inhibitory role. Therefore, high levels of miRNAs of the miR-17/92 cluster have been shown to block pro- to pre-B-cell differentiation and also advocate immature B-cell survival [29,30,31,32]. Immature B-cell survival is also advocated by miR-148a-3p. This further leads to self-reactive antibody production and subsequently B-cell elimination [32]. These data delineate the significant role of miRNAs in the primary steps of B-cell development.

### 2.2. miRNAs in Secondary Lymphoid Tissue B-Cell Development

Besides playing an important role in the development in primary lymphoid tissues, miRNAs have been reported to affect B-cell maturation in secondary lymphoid tissues as well. Specifically, a lower marginal zone B-cell number is observed upon miR-146a-5p expression, due to its binding to *NUMB* [33]. NUMB protects TP53 from degrading and advocates the Notch signaling pathway to enhance marginal zone B-cell formation [33,34,35]. On the contrary, miR-142-5p is crucial for marginal zone B-cell development, as it targets Tnfrs13c, (also known as Baff-R), which is required for B-cell maturation. Mice lacking miR-142-5p showed vigorous proliferation of B cells, due to high Tnfrs13c levels [36].

Concerning follicular B-cell maturation, plasma cell formation is the most frequently reported to be affected. More specifically, miRNAs affect the class-switch recombination, which includes further Ig gene rearrangements, leading to plasma cell formation. A relevant example is miR-181b-5p, another member of the miR-181 family; this miRNA targets *AID*, which is crucial for class-switch recombination, leading to inhibition of plasma cell formation [37,38]. In the same context, miR-125b-5p inhibits *PRDM1* and *IRF4* transcription factor expression, both of which stimulate class-switch recombination [39]. Therefore, repression of this miRNA is required during normal B-cell development; otherwise, B-cell malignancies may occur [40,41]. miR-30b-5p, miR-30d-5p, and miR-9-5p also attenuate *PRDM1* expression [42]. On the other hand, miR-148a-3p attenuates the expression of Bach2 and Mitf transcription factors, and consequently induces the expression of their downstream targets, Prdm1 and Irf4, leading to the terminal differentiation of B cells [43]. This fact highlights once again the necessity for stage-specific expression, as the expression of miR-148a-3p in early B-cell development leads to the arrest of this process [32]. In addition, miR-155-5p targets *SPI1* mRNA and hence reduces PAX5 expression levels, as *PAX5* expression is induced by the transcription factor SPI1 in both human and murine plasma cells [44]. In this way, the transition of germinal center B cells to plasma cells is advocated, as PAX5 downregulation is a necessity for that. Those miRNAs highly affecting B-cell development are presented in Figure 2.

## 3. miRNAs in Follicular Lymphoma

Follicular lymphoma (FL) is one of the most common types of NHLs deriving from B cells, as aforementioned. It usually is an indolent lymphoma; however, there is the possibility to transform into an aggressive type, namely diffuse large B-cell lymphoma (DLBCL) [45,46].

FL is a broad and extremely complex clinical entity. Many genes and cellular pathways participate in the emergence and transformation of FL. In the majority of affected tissues, a t(14;18) chromosomal translocation occurs, placing *BCL2* locus next to the immunoglobulin heavy-chain enhancer and resulting in the constitutive expression of this anti-apoptotic protein [47]. However, FL development requires the acquisition of additional aberrations that enable proliferation, immune evasion, and support from microenvironmental factors. This is usually achieved by acquired aberrations in genes that control normal germinal center B-cell development.

Precisely, in the early stages of development, FL cells acquire aberrations that enable them to (a) persist in germinal centers; (b) increase BCR signaling; (c) confer a “sustainable” level of genomic instability; and (d) inhibit apoptosis. These characteristics are achieved through mutations occurring in a set of genes (*KMT2D*, *CREBBP*, *TNFRSF14*, *EZH2*, *RRAGC*). However, these FL cells usually resemble centrocytes and, similar to their normal counterparts, have a relatively low level of proliferation. The acquisition of aberrations that enable rapid proliferation, including MYC, FOXO1, BCL6, and the BCR, TLR, and TP53 pathways, alters the tumor nature, frequently leading to histological transformation. Particularly, mutations and/or translocations in the *BCL6* genomic locus are quite important in B-cell lymphomas, since BCL6 is a transcription repressor targeting many genes, including *PRDM1*, *TP53*, *CDKN1A* and *BCL2*, thus controlling the germinal B-cell formation, cell cycle, and differentiation [48,49].

Thus far, none of the current scoring systems and therapeutic approaches have been able to mitigate the risk of early progression or histologic transformation to DLBCL. Therefore, the discovery of novel biomarkers is of significant importance.

### 3.1. miRNAs as Potential Regulators and Biomarkers in FL

Several studies support that miRNA expression profiles can serve as signatures to differentiate between different FL subtypes, which express distinctive genes and molecular markers. Different FL subtypes can divergently progress to an aggressive type. One FL subtype which has not been well-studied is t(14;18)–negative FLs, and hence the molecular events triggering FL development in cases without a t(14;18) chromosomal translocation and without high expression of *BCL2* remain largely unknown. An interesting study analyzing t(14;18)–negative FL patients and t(14;18)–positive FL patients showed that miRNA expression between these subtypes was different. Additionally, this distinct miRNA expression was reflected in the expression of their mRNA targets. One of the miRNAs with the most robust expression changes in its potential targets was miR-16-5p. More specifically, miR-16-5p was significantly decreased in t(14;18)–negative FL patients. The decreased expression levels of this miRNA were also observed in chronic lymphocytic leukemia (CLL) patients compared to non-cancerous individuals [50]. miR-16-5p has been associated with repression of the expression of *BCL2* and hence induction of apoptosis. Although in t(14;18)–negative FL patients miR-16-5p expression is also associated with apoptosis, miR-16-5p exerts its role via an alternative regulatory network. More specifically, the decreased expression levels of this miRNA lead to increased expression of its target genes *CHEK1*, which encodes an apoptosis inhibitor and DNA repair monitor, and *CDK6*, which encodes a cyclin-dependent kinase and promoter of the cell cycle (Figure 3) [48]. These findings suggest a potential mechanism which could contribute to the pro-proliferative phenotype of t(14;18)–negative FLs.

Despite the extensive investigation concerning the *BCL2* translocation in FL, *BCL6* translocation is not well-studied. Interestingly, Gebauer et al. attempted to find unique miRNA signatures between typical FL with translocation in *BCL2*, but not in *BCL6* (BCL2+/BCL6− FL) and FL with translocation in *BCL6*, but with or without translocation in *BCL2* (BCL2+/BCL6+ or BCL2–/BCL6+ FL). More specifically, in BCL2+/BCL6+ and BCL2−/BCL6+ FL patients, 21 miRNAs were significantly upregulated, and 12 miRNAs were significantly downregulated compared to BCL6− ones. Even though the functional role of these miRNAs was not further investigated and future validation of these results in a larger patient cohort is required, these results underline the differential molecular background of these subtypes and pave the way for the potential integration of miRNA signatures in FL molecular classification [51].

Additionally, the current system of differential diagnosis between distinct types of B-cell NHLs is not effective, and novel biomarkers are required. miRNAs, due to the plethora of advantages by which are characterized, have emerged as promising candidates for this aim. Especially, the discrimination between DLBCL and FL lymphoma is quite significant, since DLBCL shows high phenotypic diversity and de novo DLBCL is not easily distinguished from transformed FL [52]. Two independent analyses examined the expression pattern of miRNAs in these two different types of B-cell lymphoma; however, they resulted in distinct molecular signatures. More specifically, in the first research study, miR-200c-3p and miR-638 along with members of the miR-17/92 cluster were among the most highly expressed miRNAs in DLBCL compared to FL [53]. The second study led to different results, with limited coverage with the miRNAs of the first research study. However, this study also designated the distinct expression levels of miRNAs of the miR-17/92 cluster [54].

Particularly, the miR-17/92 cluster (genomic locus: 13q31.3) and its two paralogs, namely the miR-106b-25 and miR-106a-363 clusters, have been reported to be involved in several hematological malignancies, and more specifically in the most aggressive ones. This could be partly attributed to enhanced transcription of the miR-17/92 cluster host gene (*MIR17HG*) by MYC oncoprotein and, thus, the upregulation of 6 oncogenic miRNAs (miR-17-5p, miR-18a-5p, miR-19a-3p, miR-19b-3p, miR-20a-5p, and miR-92a-3p). miR-17-5p inactivates CDKN1A, leading to deregulation of the cell cycle and increased cell proliferation. miR-18a-5p and miR-19a-3p repress CD32 (Fc fragment of IgG receptor IIb, FCGR2B) and CD22, respectively, resulting in the upregulation of the BCR signaling pathway, and hence in elevated B-cell activation and division [55]. Moreover, miR-19b-3p inhibits PTEN phosphatase [56], a regulator of cell cycle and growth, which subsequently suppresses the oncogenic PI3K/AKT signaling pathway [57,58]. Considering all these findings, further investigation of this miRNA cluster role in FL is considered fruitful.

An additional miRNA that is implicated in the high-grade transformation of FL is miR-150-5p. This miRNA plays, also, a key role in normal B-cell development via targeting *MYB* [14]. In FL, its expression is repressed by MYC, leading to its decreased expression levels in FL cells and, consequently, high expression levels of another one of its targets, the transcription factor *FOXP1*. The elevated expression of the latter has been linked to lower survival rates and high-grade malignant transformation. This could be attributed to its role, since FOXP1 regulates the expression of many genes involved in cell survival and cell cycle activation, and promotes BCR signaling, while it is also critical for normal B-cell development [57,59].

Furthermore, in a patient cohort study it has been observed that as the disease progresses from FL to aggressive DLBCL, miR-31-5p expression levels decrease. This miRNA has attracted researchers’ interest due to its multifaceted role. Depending on its specific targets in distinct cell types, miR-31-5p can exert either an oncogenic or an onco-suppressive role in several malignant states. The low expression of this miRNA has also been observed in a cohort study with DLBCL patients [60]. This low expression can be achieved either by loss of the gene locus of this miRNA or by hypermethylation of its promoter, while both mechanisms have been detected in different malignancies. The *MIR31* gene is located on chromosome band 9p21.3, ∼500  kb from the locus of the well-known tumor suppressors CDKN2A and CDKN2B. Due to their proximity, it is reasonable to suppose that *MIR31* would be lost together with *CDKN2A* [61]. The deletion of the latter has been associated with poor prognosis of DLBCL patients. In a recent study regarding FL transformation, it was observed that *E2F2* and *PIK3C2A* could be direct targets of miR-31-5p. E2F2 is a transcription factor of the E2F family, which permits the entry of cells to the S-phase of the cell cycle, thus promoting the cell cycle, and PIK3C2A, which is a catalytic subunit of the PI3K family, is involved in cell migration, survival, and proliferation. High levels of E2F2 and elevated activity of the PI3K/AKT signaling pathway have been observed in DLBCL, while the latter has been associated with poor outcome of DLBCL patients, as well. Therefore, low miR-31-5p expression levels could result in a B-cell high-grade tumor, via the increased levels of the aforementioned proteins [62]. Additionally, the same study uncovered the increase of the expression levels of miR-17-5p during FL transformation, a finding which is consistent with the oncogenic role of this miRNA in several other malignancies.

Besides miRNA signatures with the potential to discriminate between different B-cell NHLs, an interesting study revealed a miRNA signature capable of distinguishing FL cells from normal germinal center B cells in follicular hyperplasia. The most highly expressed miRNAs in FL included miR-20a-5p, miR-20b-5p, and miR-194-5p. The first two miRNAs have been proved to target *CDKN1A*, which partly accounts for cell cycle arrest, while miR-194-5p controls the expression of SOCS2, a suppressor of the JAK/STAT signaling pathway, which participates in cell proliferation and survival [63].

miRNAs exerting a regulatory role in indolent B-cell NHLs, along with their targets and effect in malignant B cells, are summarized in Table 1, while Table 2 highlights those miRNAs showing a potential clinical utility as candidate biomarkers in FL.

### 3.2. Genetic Polymorphisms of miRNA Genes in FL

Genetic variation in miRNA regulatory pathways in distinct malignancies has raised researchers’ interest, as well. These polymorphisms can be developed in miRNA-binding site target genes, in miRNA biogenesis pathway genes, and in different regions in miRNA genes. Therefore, they can change the function of the respective miRNA and serve as potential indicators for diagnosis and prognosis in clinical practice. Although these miRNA variants or single nucleotide polymorphisms (SNPs) have been extensively examined in a wide range of malignancies, the knowledge of miRNA SNPs in FL and generally in B-cell NHLs remains poor [84].

miR-202-3p and miR-618 have been implicated in FL, while the SNPs in their precursor sequences are associated with FL, via the impact on the levels of the target gene expression. More specifically, the presence of these miRNA SNPs has been linked with elevated risk for FL. A possible explanation of this finding is that the presence of a SNP in mir-202 and mir-618 could lead to a decrease in the expression levels of miR-202-3p and miR-618, respectively, which seem to exert onco-suppressive roles. miR-202-3p has been shown to target *DICER1*, which is essential for the proper biogenesis and function of all miRNAs, while its high expression has been associated with B-cell lymphoma development and survival. Additionally, miR-202-3p targets *SKP2*, which encodes a regulator of G1 to S-phase transition of the cell cycle and inhibitor of CDKN1B [64]. Concerning miR-618, it targets genes encoding histone deacetylase, HDAC3, which represses the function of the onco-suppressor, TP53, and CUL4A, a protein that is involved in the degradation of DNA damage-response proteins, TP53 and TP73 [65].

### 3.3. miRNAs and the Immune System

The immune system activation plays a critical role in inhibiting the progression of nascent tumors by recognizing specific antigens on the surface of malignant cells, while it has been investigated in the context of FL as well. For instance, the natural killer (NK) cell receptor KLRK1 was found to bind both MICA and MICB, thus leading to suppression of B-cell lymphomas by inducing cell cytotoxicity. This finding is consistent with the high expression levels of the aforementioned KLRK1 ligands in low-grade FL and their low expression levels in high-grade FL. The increased expression levels of miR-93-5p in high-grade FL are associated with the lower expression levels of MICA and MICB, since the respective mRNAs constitute direct targets of miR-93-5p [67].

Immunotherapy and specifically monoclonal antibodies (mAb) have been introduced in FL treatment arsenal. Particularly, Obinutuzumab, a humanized anti-CD20 mAb, is approved as a treatment for FL. This mAb increases the affinity between the CD20 receptor of malignant B cells and the CD16 receptor of NK cells, leading to increased INF-γ levels. Interestingly, miR-155-5p is implicated in this procedure, as was shown in a recent study. More specifically, it was observed that obinutuzumab-induced CD16 stimulation led to overexpression of miR-155-5p, which targeted INPP5D inositol phosphatase, a regulator of the PI3K pathway. This resulted in the activation of the downstream target of PI3K, MTOR, and the stimulation of IFNγ production from NK cells, triggering anti-tumor immune responses [85]. The NK cells are pivotal components of innate immunosurveillance against malignancies and represent a particularly attractive tool in the context of anti-tumor immunotherapy. NK cells rapidly recognize and destroy many tumor cell types and also play an immunoregulatory role in the instruction of adaptive anti-tumor responses [86]. Therefore, deciphering the cascade following the treatment with mAb is critical, since it is expected to contribute utmost to the optimization of treatment strategies against malignancies.

**Table 2 biomedicines-09-00333-t002:** miRNAs as candidate biomarkers in follicular lymphoma (FL).

miRNAs	Sample Origin	Expression	Potential Biomarker	References
miRNAs of miR-17/92 and miR-106a-363 clusters, miR-200c-3p, miR-638, miR-518a-3p	FFPE tissues	Upregulated in DLBCL vs. FL	Differential diagnosis	[53]
miR-17-5p				[54]
miR-217-5p, miR-634, miR-26b-5p		Upregulated in FL vs. DLBCL		[53]
miR-330-3p, miR-106a-5p, miR-210-3p		Upregulated in FL vs. DLBCL		[54]
miR-612, miR-188-5p, miR-302c-3p, miR-433-3p, miR-584-5p		Upregulated in BCL2+/BCL6+ and BCL2−/BCL6+ FL vs. BCL2+/BCL6− FL		[51]
miR-200a-3p, miR-135a-5p, miR-375-3p, miR-138-5p, miR-517 isomiRs		Downregulated in BCL2+/BCL6+ and BCL2−/BCL6+ FL vs. BCL2+/BCL6− FL		
miR-16-5p, miR-138-5p, miR-26a-5p, miR-29c-3p		Downregulated in t(14;18)–negative FL vs. t(14;18)–positive FL		[48]
miR-193a-5p, miR-345-5p, miR-574-3p, miR-1287-5p, miR-1471	Enriched FL cells	Upregulated in FL vs. with follicular hyperplasia		[63]
miR-570-3p, miR-205-5p, miR-222-3p, miR-30a-5p, miR-301b-3p, miR-141-3p		Downregulated in FL vs. with follicular hyperplasia		
miR-20b-5p, miR-26a-5p, miR-92b-3p, miR-487b-3p	Cancer cell lines	Upregulated in FL cell lines vs. DLBCL cell lines		[87]
miR-330-3p, miR-106a-5p, miR-210-3p, miR-301 isomiRs, miR-338-5p	FFPE tissues	Upregulated in FL vs. non- neoplastic lymph nodes	Diagnosis	[54]
miR-149-5p, miR-139-5p		Downregulated in FL vs. non- neoplastic lymph nodes		
miR-16-5p, miR-17-5p, miR-26a-5p, miR-29a-3p, let-7d-5p, let-7g-5p, let-7i-5p	Cancer cell lines	Downregulated in FL cell lines vs. CD19+ negatively- selected B cells		[87]
miR-144-3p, miR-431-5p	FL biopsies and blood samples	Upregulated in relapsed FL patients	Prognosis, prediction of disease progression	[88]
miR-376b-3p		Downregulated in non-relapsed FL patients		

Abbreviations: DLBCL, diffuse large B-cell lymphoma; FFPE, formalin-fixed, paraffin-embedded.

## 4. miRNAs in Marginal Zone Lymphoma

### 4.1. Extranodal Marginal Zone Lymphoma or Mucosa-Associated Lymphoid Tissue (MALT) Lymphoma

Extranodal MZL, also known as MALT lymphoma, is the most common type of indolent MZL and it starts at places where malignant marginal zone B cells initially infiltrate MALTs, other than the lymph nodes (hence the name extranodal). Stomach is the most common organ where this malignancy can arise, accounting for almost half the incidences. This type of lymphoma is known as MALT gastric lymphoma (MALT GL). Less frequently, MALT lymphoma can start at organs other than the stomach (non-gastric), such as the skin. The exact underlying mechanisms of this disease are not yet known, but clinical and epidemiological data have profoundly associated the high risk of MALT lymphoma development with certain chronic infections and autoimmune diseases. The *Helicobacter pylori* infection and Sjögren’s syndrome constitute the most important leading factors, respectively [89,90,91,92].

### 4.2. Gastric MALT Lymphoma and H. pylori

A major leading factor of MALT GL is chronic inflammation triggered by persistent *H. pylori* infection. As a proof, *H. pylori* eradication can fully treat MALT GL in a large number of cases [90,93]. The majority of patients with MALT GL resistant to *H. pylori* eradication therapy, have in common a t(11;18)(q21;q21) chromosomal translocation [94]. This translocation leads to the production of a fusion gene, consisting of the apoptosis inhibitor *BIRC3* and the caspase-like protease *MALT1*. The encoded fusion protein enhances NFkB signaling and hence leads to inhibition of apoptosis [95,96,97].

Saito et al. contributed greatly to the elucidation of the mechanisms involved in MALT GL progression. Firstly, they indicated that miR-142-5p and miR-155-5p are overexpressed in MALT GL compared to non-tumor gastric mucosa, while their levels were significantly higher in patients unresponsive to *H. pylori* eradication treatment, compared to the responsive ones, implying their potential prognostic and predictive utility as biomarkers. Considering the aforementioned critical role of these two miRNAs in normal B-cell development, their further investigation is quite significant. Interestingly, two of the patients in this study were resistant to *H. pylori* eradication therapy but lacked the *BIRC3-MALT1* fusion gene. Nevertheless, they were also characterized by increased expression of miR-142-5p and miR-155-5p, suggesting that these molecules could be used as additional biomarkers in MALT GL [75]. Moreover, it was shown that they target the pro-apoptotic gene *TP53INP1* (Figure 4). This interaction could lead to inhibition of apoptosis and acceleration of MALT GL cell proliferation, designating miR-142-5p and miR-155-5p as potential therapeutic targets [75]. Additionally, a former research study had revealed that *TP53INP1* transcription is activated by TP73, and therefore cell cycle arrest is facilitated [98]. Considering this additional regulation level of *TP53INP1*, it would be interesting to be further investigated in the context of miR-142-5p and miR-155-5p expression.

Another relevant study concluded that miR-383-5p is downregulated in MALT GL patients infected with *H. pylori*, compared to normal non-tumor gastric mucosae tissues, and determined *ZEB2* as its target [72]. High levels of ZEB2 have previously been reported to promote epithelial-mesenchymal transition (EMT) of gastric cancer cells via regulation of expression of CDH1 (E-cadherin) and other EMT markers, such as VIM (vimentin) and matrix metallopeptidases (MMP2 and MMP9) [99]. Thus, the effect of reduced expression of miR-383-5p and the high expression of *ZEB2* could assist in the understanding of the role of *H. pylori* infection in MALT GL development.

### 4.3. From Chronic Gastritis to MALT GL

Chronic gastritis can also progress to MALT GL development; however, the molecular basis of this transformation remains unknown. Craig et al. identified miR-203a-3p to be significantly underexpressed in MALT GL compared to normal lymphoid tissue. Furthermore, *MIR203A* promoter was hypermethylated, and its target protein, ABL1, was overexpressed in MALT GL in comparison with gastritis, indicating ABL inhibitors as a novel therapeutic approach in MALT GL [73]. ABL1 is a tyrosine kinase that activates various signaling pathways, including BCR, leading to cell proliferation. The BCR signaling activity is elevated in several hematological malignancies, while its targeting constitutes a therapeutic approach in several cancers. In accordance with the aforementioned observations regarding miR-155-5p overexpression in *H. pylori*–positive MALT GL patients [75] and the miR-203a-3p underexpression in MALT GL, findings from an independent study showed a decrease in miR-203a-3p expression levels and a concomitant increase in miR-155-5p expression levels in MALT GL patients, compared to chronic gastritis patients [74]. The same study designated the high expression levels of miR-142-3p in MALT GL compared to chronic gastritis (Table 3). These findings are quite important since the morphological diagnosis of MALT GL is still hampered by overlapping histological features with advanced chronic gastritis. Considering that either the 3′ or the 5′ of a miRNA stem-loop is expressed under certain circumstances, the overexpression of miR-142-5p and miR-142-3p in MALT GL patients would be interesting to examine further.

The role of miRNAs in malignant transformation from chronic gastritis to MALT GL has been investigated in other studies, as well. Another miRNA with a critical role in normal B-cell development, via targeting the transcription factor *MYB* [14], and with deregulated expression levels in MALT GL, is miR-150-5p. More precisely, it was found to be significantly overexpressed in MALT GL in comparison with chronic gastritis [76,100]. On the contrary, miR-150-5p expression levels were low during FL transformation [59]. Considering these findings, it could be fruitful to investigate the functional role of miR-150-5p in MALT GL. Additionally, findings deriving from gastric cancer research revealed that miR-150-5p inhibits apoptosis in gastric cancer cells by targeting the pro-apoptotic gene *EGR2* [101]. This molecule has been investigated in hematological malignancies and its deregulated expression or potential mutations have been associated with tumorigenesis [102,103]. Therefore, this could be a potential mechanism of action via which miR-150-5p could exert its role in MALT GL.

Additionally, Blosse et al. showed that miR-150-5p, miR-155-5p, miR-196a-5p, and miR-138-5p were upregulated and miR-7-5p and miR-153-3p were downregulated in MALT GL patients compared to gastritis control patients [76]. miR-150-5p and miR-155-5p have been associated with MALT GL in other studies, as aforementioned. The convergence of several research studies in the deregulated expression of these two miRNAs highlights their critical role in MALT GL. Regarding the deregulated expression levels of the rest miRNAs and their significant role in other malignancies, the investigation of their function in MALT GL is critical, as well. Herein, we propose potential mechanisms of action of these miRNAs, based on current literature. Interestingly, miR-196a-5p is highly expressed in gastric cancer cells and targets the cell cycle inhibitor *CDKN1B* (*p27^kip1^*), leading to increased cell proliferation [104], while miR-153-3p acts on *AKT3* in lung cancer reducing cell proliferation rate [105]. Moreover, *EGFR* and *IGF1R* have been proposed as targets of miR-7-5p in gastric cancer cells, suggesting a way that miR-7-5p can suppress the invasion and metastasis of these malignant cells [106,107]. Finally, conflicting studies have been made about the role of miR-138-5p in gastric cancer cells, as some support its role as onco-suppressive or as oncogenic miRNA [108,109], so further investigation is required to properly decipher its role.

### 4.4. From MALT GL to Gastric DLBCL

An alarming situation arises when MALT GL is transformed to gastric DLBCL through mechanisms which are not well understood. A microarray analysis between these two pathological states revealed 27 underexpressed miRNAs in gastric DLBCL compared to MALT GL [68]. These miRNAs were transcriptionally repressed by MYC, as previously shown in a B-cell lymphoma mouse model [69], while miR-34a-5p possessed the most tumor-suppressive properties [68]. In the same study, MYC was found to be greatly overexpressed in gastric DLBCL in comparison with MALT GL, and the MIR34A promoter was also found to be hypermethylated only in gastric DLBCL. Moreover, the validated target of miR-34a-5p in the aforementioned study was *FOXP1* [68]. During normal B-cell development, constitutive expression of miR-34a-5p can result in a block in B cell development at the pro-B cell to pre-B cell transition, leading to a reduction in mature B cells. This block appeared to be mediated primarily by inhibited expression of the *FOXP1* [110]. Several previous studies highlighted that FOXP1 is necessary for normal B-cell differentiation [22] and has been reported to predict the transformation of MALT GL to gastric DLBCL [70]. Indeed, data from another clinical study confirm that miR-34a-5p could be utilized as a prognostic biomarker to investigate MALT GL to gastric DLBCL transformation [71]. These results suggest a novel way that FOXP1 can lead to MALT GL progression, besides the t(3;14)(p14;q32) chromosomal translocation that results in a *IGH-FOXP1* fusion gene and, therefore, in elevated levels of FOXP1 in MALT lymphomas [111,112]. Moreover, FOXP1 was found to be elevated in high-grade lymphomas resulting from transformation of FL [59]. All these findings highlight the pivotal role of *FOXP1* in the development of B-cell malignancies and hence the role of miR-34a-5p as one of its potential regulators.

In addition, Gu et al. demonstrated that miR-16-5p had higher expression levels in MALT GL patients than those with gastric DLBCL and could be used as another biomarker predicting MALT GL transformation [113]. miR-16-5p is a key tumor-suppressive miRNA and has repeatedly been associated with CLL and, as aforementioned, with FL. Its more well-known target is the anti-apoptotic gene *BCL2*, via the suppression of which can inhibit apoptosis in CLL [114]. Even though its potential functional role in MALT GL transformation has not been unraveled, the existing data regarding its function in other hematological malignancies highlight its further investigation in this cancer, as well.

### 4.5. Non-Gastric MALT Lymphoma

Studies investigating the role of miRNAs in non-gastric MALT lymphomas are far less frequent. Cutaneous marginal zone B-cell lymphoma is another extranodal MZL, which has been associated with *Borrelia burgdorferi* infection [115,116]. Reduced expression of miR-150-5p and miR-155-5p in primary cutaneous MZL in comparison with primary cutaneous centrofollicular lymphoma has been linked to disease deterioration and lower survival rates only in primary cutaneous MZL [117]. Therefore, these two miRNAs could be used to predict the outcome of this lymphoma type. Comparing the aforementioned results which propose that miR-150-5p and miR-155-5p are elevated in gastric MALT lymphoma and could contribute to this lymphoma progression [75,76,100] and the results regarding the expression levels of these two miRNAs in primary cutaneous MZL, a conflict in their role emerges. Probably this could be attributed to the involvement of these miRNAs in several networks only certain of which could prevail depending on the organ and the microenvironment (stomach or skin). The fact that the expression levels of miR-150-5p were decreased during FL transformation supports the aforementioned hypothesis [59]. However, further investigation is needed.

Ocular adnexal lymphoma (OAL) is another less common type of extranodal MZL affecting tissues surrounding the eye, though the driving mechanisms of this disease are still under investigation. The only study to date to have examined the miRNA expression profiles in OAL has revealed that let-7g-5p, miR-16-5p, members of the miR-29 family, miR-199a-5p, and miR-222-3p were overexpressed in OAL in comparison with ocular DLBCL, the aggressive transformed malignancy which can arise in some patients [118]. Strikingly, transcription of many of these miRNAs is suppressed by MYC, which usually drives B-cell proliferation [69]. Thus, transcriptional repression of miRNA host genes, mediated by MYC, most likely contributes to the transformation of OAL to ocular DLBCL. The majority of these miRNAs play a critical role in hematological malignancies and solid tumors, necessitating their further investigation in B-cell NHLs, as well. Below, some potential modes of action of these miRNAs are proposed. One of the most critical miRNAs for further investigation is miR-16-5p, as it has been repeatedly characterized as a pivotal tumor suppressor in B-cell malignancies [113]. Furthermore, the members of the let-7 family can act as regulators of stem-cell differentiation and have also been implicated in tumor suppression in several ways. Interestingly, some members of this family suppress the acquisition and utilization of key nutrients, which are essential for B-cell activation. Additionally, members of the miR-29 family have been characterized as tumor suppressors in other malignancies, including mantle cell lymphoma, Burkitt lymphoma, and FL. This miRNA family is implicated in the regulation of several key pathways in carcinogenesis. Some of its main target genes are *CDK6*, *DNMT3B*, *TCL1A*, and *MCL1*, which are involved in cell cycle control, DNA methylation, and apoptosis inhibition, respectively [119]. Regarding miR-199a-5p, its high expression has been associated with a better outcome in DLBCL patients, while one of its potential roles is the suppression of the NFkB signaling pathway, a critical pathway for the development of this malignancy [120,121]. Finally, miR-222-3p is another miRNA with a contradictory role since it has been characterized both as oncomiR and a tumor suppressor in lymphomas, highlighting the complex regulatory roles and networks of miRNAs.

Besides chronic bacterial infections, Sjögren’s syndrome (SS) is a chronic autoimmune disease affecting predominantly exocrine glands, in which a considerable percentage of patients are at high risk of developing B-cell NHL, with MALT lymphoma being the most frequent subtype [91,122]. miR-200b-5p was found to be significantly underexpressed in minor salivary glands of SS patients with MALT lymphoma compared to SS patients without lymphoma [123]. Interestingly, in another study, it has been revealed that low expression levels of this miRNA in minor salivary glands could predict SS patients who are at high risk of B-cell NHL development, even before the appearance of clinical symptoms of the disease. However, these results necessitate validation in a larger cohort of patients [124]. The discovery of a potential implication of this miRNA in MALT lymphoma is quite important since until recently it was believed that it was degraded and the miR-200b-3p prevailed. However, recent data support the synergistic action of both miRNAs in the inhibition of EMT [125]. Therefore, the functional investigation of miR-200b-5p in MALT lymphoma and B-cell development in general could be interesting.

### 4.6. Splenic Marginal Zone Lymphoma

Splenic marginal zone lymphoma (SMZL) is an indolent B-cell NHL, but the possibility to be transformed eventually to more aggressive type lymphomas is quite high. It originates in the spleen and lacks a clear etiology, which necessitates the discovery of the implicated molecular mechanisms and of indicators for disease development and progression. Due to the multifaceted role of miRNAs in normal B-cell development, miRNAs can prove to be beneficial in the aforementioned endeavor [126]. The most common chromosomal abnormality, present in approximately 40% of SMZL cases, is a characteristic 2.8-Mbp 7q32 heterozygous deletion [127,128]. This genomic locus comprises the host genes of several miRNAs that are underexpressed in SMZL, including *MIR593*, *MIR129-1*, *MIR182*, *MIR96*, *MIR183*, *MIR335*, *MIR29A*, and *MIR29B1* [128]. These miRNAs are involved in various regulatory networks affecting cell differentiation, proliferation, and apoptosis, and hence their specific role in SMZL requires further investigation. However, indicative mechanisms of action have been proposed for the most significantly underexpressed miRNAs, mainly in other types of malignancies [126]. In detail, miRNAs of the miR-29a/b1 cluster are believed to act as tumor suppressors by inhibiting the expression of the *TCL1A* oncogene, one of the most overexpressed genes in SMZL [129]. This interaction has already been observed in CLL [130]. Additionally, miR-129-5p has been shown to target the notorious anti-apoptotic *BCL2* mRNA in colorectal cancer, both in vitro and in vivo, leading to apoptosis and enhancing the cytotoxic effect of 5-fluorouracil [131]. On the contrary, miR-182-5p and miR-183-5p have been characterized as oncogenic miRNAs. More specifically, their overexpression in mesothelioma leads to enhanced cell proliferation and invasion. This function is established mainly by preventing the expression of *FOXO1* transcription factor, which in turn facilitates the expression of *CDKN1B*, a key inhibitor of CDKs [132]. Therefore, their downregulation in the present research study raises questions regarding their function.

Another study showed that miR-21-5p overexpression in SMZL is linked to an aggressive transformation type of the disease [133]. Even though the role of miR-21-5p has not been investigated in SMZL, there are several studies that characterize it as oncomiR in NHLs, while there are studies, which examine this miRNA as a therapeutic target. More precisely, it inhibits the expression of PTEN and FOXO3, which are molecules with a critical role in normal B-cell development, in human B-cell NHL cell lines. This inhibition activates the PI3K/AKT pathway and renders a human DLBCL cell line resistant to chemotherapy [134].

Studies demonstrate that hepatitis C virus (HCV) infection is a risk factor for SMZL development, but the underlying mechanisms leading to this condition are poorly understood. A solid argument supporting this notion is that HCV-positive SMZL patients who received antiviral treatment achieved complete or partial remission [135,136,137]. An extensive miRNA profiling in SMZL revealed five miRNAs to be overexpressed and seven miRNAs having decreased expression in SMZL compared to the non-tumor splenic marginal zone. Following the stratification of SMZL patients to HCV-positive and -negative ones, miR-26b-5p was proved to be significantly underexpressed in HCV-positive SMZL patients compared to negative ones, and *NEK6* is a predicted target of miR-26b-5p [77]. NEK kinases, including NEK6, facilitate many mitotic events and, subsequently, cell division, while they are critical for STAT3 phosphorylation and hence JAK/STAT signaling pathway activation [78]. Although the overexpression of *NEK6* has not been associated with SMZL, it has been associated with the development of other malignancies [138]. Thus, the decreased expression of miR-26b-5p and the subsequently increased expression of *NEK6* in HCV-positive patients suggest a molecular mechanism of action through which HCV infection could lead to SMZL.

### 4.7. Nodal Marginal Zone Lymphoma

Nodal marginal zone lymphoma (NMZL) is a rare MZL subtype initiating in the lymph nodes which is challenging to differentiate diagnostically, due to the lack of specific indicators for it. Considering the decisive role of miRNAs in B-cell developmental stages, potential distinct expression patterns of these regulatory molecules could serve as useful biomarkers [139,140]. Intriguingly, Arribas et al. conducted a miRNA expression analysis accompanied by an analysis of their targets, in order to conclude in a miRNA signature able to distinguish NMZL from FL [139]. There are some cases of NMZL that are challenging to be distinguished and a combination of clinical, histological, immunohistochemical, and molecular data is required, so these findings could be an additional tool for classifying patients standing in the diagnostic grey zone [141]. Although further investigation is required, they showed that miR-223-3p and let-7f-5p were the most highly expressed miRNAs in NMZL compared to FL. Interestingly, miR-223-3p has been proved to regulate naïve to germinal center B-cell transition and germinal center to memory B-cell transition, via the repression of the key protein for hematopoietic development, LMO2 [42]. FL and germinal center cells are distinguished by an increased expression of LMO2, and a diminished expression of miR-223-3p. In the aforementioned study, it was shown that the expression of LMO2 was low and the expression of miR-223-3p was high in NMZL patients, implicating a potential role of these molecules in NMZL development. Although it is known that let-7f-5p is a member of the let-7 miRNA family, which has been shown to target various oncogenes and is usually underexpressed in many malignancies [142], a functional explanation of the differential expression of this miRNA among these malignancies is not provided. Another study, conducted by Gebauer et al., identified several miRNAs differentially expressed between transformed NMZL, which is characterized by the presence of larger cells under histopathological examination, and DLBCL. These distinct miRNA signatures support the notion that transformed NMZL is biologically a distinct disease entity, while the presence of large cells in some cases of NMZL does not correspond to an aggressive type transformation into DLBCL [143,144].

**Table 3 biomedicines-09-00333-t003:** miRNAs as candidate biomarkers in marginal zone lymphoma (MZL).

Disease	miRNAs	Sample Origin	Expression	Potential Biomarker	References
Gastric MALT lymphoma	miR-142-3p, miR-155-5p	FFPE tissues	Upregulated in gastric MALT lymphoma vs. chronic gastritis	Differential diagnosis	[74]
	miR-203a-3p		Downregulated in gastric MALT lymphoma vs. chronic gastritis		
Ocular adnexal lymphoma (OAL)	let-7g-5p, miR-16-5p, miR-29 family, miR-199a-5p, miR-222-3p	FFPE tissues	Upregulated in OAL vs. ocular DLBCL	Differential diagnosis	[118]
Sjögren’s syndrome (SS) associated with MALT lymphoma	miR-200b-5p	Minor salivary glands and PBMCs	Downregulated in SS-associated MALT lymphoma vs. SS	Prognosis, prediction of patients’ relapse	[123,124]
Splenic MZL (SMZL)	miR-21-5p	Fresh frozen and FFPE tissues	Upregulated in aggressive SMZL vs. indolent SMZL	Prognosis, prediction of patients’ relapse	[133]
Nodal MZL (NMZL)	miR-223-3p, let-7f-5p	FFPE tissues	Upregulated in NMZL vs. FL	Differential diagnosis	[139]

Abbreviations: DLBCL, diffuse large B-cell lymphoma; FFPE, formalin-fixed, paraffin-embedded; MALT, mucosa-associated lymphoid tissue; PBMCs, peripheral blood mononuclear cells.

## 5. miRNAs in Rare Types of Indolent B-Cell NHLs

### 5.1. Waldenström’s Macroglobulinemia or Lymphoplasmacytic Lymphoma

Waldenström’s macroglobulinemia (WM) or lymphoplasmacytic lymphoma is an indolent B-cell NHL, characterized by an arrest of B cells after somatic hypermutation and prior to isotype class switching. These cancer cells produce large amounts of immunoglobulin M (IgM). The accumulation of these malignant cells can result in an indirect reduction of red and white blood cells in the bone marrow, leading potentially to hyperviscosity due to the IgM aggregation, anemia, and to attenuated functionality of the immune system.

A common genetic alteration in patients with WM, which is present in more than 90% of patients, is a mutation in the *MYD88* gene; this mutation is also abundant in other B-cell malignancies with a different frequency. In WM, this mutation leads to an activation of the NFkB signaling pathway, resulting in the growth and survival of the WM cancer cells. Considering the highly regulatory potential of miRNAs, their role in this pathway could be critical for WM. More specifically, miR-23b-3p was found downregulated in patients with WM, and transfection experiments with miR-23b-3p mimics resulted in a decrease in WM cell proliferation and survival. In a functional experiment, miR-23b-3p was found to target *SP1* 3′ UTR, which is an overly activated transcriptional factor positively affecting the NFkB signaling pathway in WM and multiple myeloma. Moreover, the *MIR23B* promoter was found to be under transcriptional control by MYC. The downregulation of MYC resulted in increased levels of miR-23b-3p, proposing an important MYC/miR-23b-3p/SP1 regulatory axis with a significant role in the proliferation and survival of WM cells [80]. Furthermore, another genetic alteration has been found in cells of WM patients, associating with miRNA functions. Specifically, a study showed a deletion in the 13q14 chromosomal region in 10% of WM patients [145]. This region includes the genomic location from which miR-15a-5p and miR-16-5p are generated. These two miRNAs which possess a critical regulatory role in other malignancies such as CLL, downregulate BCL2 anti-apoptotic protein leading to apoptosis of cancer cells. Moreover, these two miRNAs have been characterized as negative regulators of the NFkB signaling pathway. As a result, the downregulation of these two miRNAs in WM patients could lead to increased proliferation and survival of the malignant cells. All the aforementioned information shows a downregulation of these miRNAs in WM cells, leading to activation of NFkB signaling and enhanced cancer cell properties.

Another important miRNA in WM is miR-155-5p, which has also a significant impact on other B-cell malignancies and normal B-cell development. In WM, this miRNA was found upregulated in comparison with cells from healthy individuals. Functional studies, in which miR-155-5p was knocked down, found a critical involvement of this miRNA in the PI3K/AKT and the NFkB signaling pathways in WM. Moreover, miR-155-5p was found to dispose a positive regulation in the proliferation, adhesion, and migration of WM cells [81]. Furthermore, in WM cells with augmented miR-155-5p expression, a significant downregulation of FOXO3 transcription factor and BCL2L11 pro-apoptotic BCL2 family member was observed, resulting in abrogation of the activation of apoptosis [83]. Additionally, another study enforced the knowledge about the impact of this miRNA in WM. More specifically, in experiments where WM cells with knocked-down miR-155-5p were treated with everolimus, an MTOR inhibitor, inhibition of cytotoxicity was observed, in comparison with WM cells with normal levels of miR-155-5p [82]. Furthermore, in another experiment, the levels of miR-155-5p were found downregulated in a dose-dependent manner as everolimus concentration augmented. All the aforementioned information highlights the significance of miR-155-5p in WM, not only with regard to the way it affects important pathways and cell properties, but its therapeutic potency, as well.

miRNAs have also been reported as potential biomarkers for WM (Table 4) [81,83], with a part of them also possessing a regulatory role in this malignancy, as summarized in Table 1. After the observation that miR-206-3p expression levels are elevated and miR-9-3p expression levels are lower in WM cells compared to cells from normal individuals, a study showed a change in the levels of histone deacetylases (HDACs) and histone acetyltransferases (HATs) in WM cells, following the downregulation of miR-206-3p and the upregulation of the miR-9-3p [79]. Specifically, miR-206-3p was found to downregulate the histone acetyltransferase KAT6A and miR-9-3p was found to downregulate the histone deacetylases HDAC4 and HDAC5. This epigenetic regulation which is driven by miRNAs is of high importance, as deregulation of HDACs and HATs is a common phenomenon in numerous malignancies. However, further research is essential to shed more light on the role of miRNAs in this malignancy, as current knowledge remains limited.

### 5.2. Other Rare Types of B-Cell NHLs

Hairy cell leukemia (HCL) is a rare type of leukemia with an incidence of 0.3/100,000 people. The malignant cells are a type of B lymphocyte, but they’re different from those seen in CLL. They possess projections coming off them that give them a “hairy” appearance, hence the name of the disease. Specific miRNAs have been identified as biomarkers in this malignancy, as well. In 2011, Moussay et al. found a downregulation in the levels of circulating miR-363-3p and miR-708-5p in plasma samples of patients with HCL in comparison with patients with CLL [146]. This observation may be of high importance for the discrimination between these two malignancies. In another study, six miRNAs were found upregulated in patients with HCL in comparison with normal samples or patients with other malignant B cells [147]. Target prediction of these six molecules revealed the regulatory impact on MAPK pathways, mainly via targeting of the molecules which contribute to the activation of the JNK signaling pathway, which has an apoptotic effect in HCL. This negative regulation of the pro-apoptotic JNK signaling pathway may possess an anti-apoptotic effect in HCL cells, leading to prolonged survival of the cancer cells.

miRNAs also appear as promising molecules in PCFCL, another type of indolent B-cell NHLs, with limited information regarding its pathogenesis. As previously described in this review, Monsalvez et al. uncovered the differential expression of miR-150-5p in comparison with primary cutaneous marginal zone B-cell lymphoma. This miRNA, which targets the transcription factor MYB, a factor that participates in the proliferation and differentiation of hematopoietic progenitor cells, has been found downregulated in PCFCL [117]. Moreover, the fact that this miRNA is differentially expressed in other types of indolent B-cell NHLs points out its high significance in these malignancies. Additionally, another study comparing PCFCL and primary cutaneous DLBCL-leg type, highlighted four other miRNAs (miR-9-5p, miR-31-5p, miR-129-2-3p, and miR-214-3p) which could be used for the distinction between these two malignancies [148]. These miRNAs have been acknowledged both as tumor-suppressors and as oncogenes in distinct malignancies. Moreover, miR-9-5p, miR-31-5p, and miR-214-3p were found to regulate the activity of signaling pathways such as the NFkB and PI3K/AKT.

Specific miRNAs with potential biomarker utility in the aforementioned rare types of indolent B-cell NHLs are summarized in Table 4. The identification of specific miRNA signatures with biomarker utility that can be used in order to distinguish specific rare types of indolent B-cell NHLs from other types of leukemia, is of high importance for the timely and optimal management of patients. Moreover, discovering miRNAs with biomarker utility may reveal other promising molecules.

Although all the aforementioned information is promising, further research is essential to elucidate the involvement of miRNAs in rare types of indolent B-cell NHLs. Elucidating the regulatory effect of miRNAs with different levels in rare indolent B-cell NHLs may reveal novel candidates that participate in pathogenic events that lead to these distinct malignancies. Characteristically, as previously mentioned, in all three types of rare indolent B-cell NHLs which are presented in this review, there are paradigms of miRNAs, having a potential biomarker utility, which can also have an oncogenic or an onco-suppressive role in each disease. Moreover, predicted targets of miR-363-3p, miR-494-3p, miR-184-3p, and miR-542-3p, which are increased in WM patients include tumor suppressors, cell-cycle inhibitors, cytokine signaling suppressors, and tyrosine phosphatases [81]. miR-9-3p, which acts as a onco-suppressor and is decreased in WM patients, targets protein kinases, oncogenes, and transcription factors enhancing apoptosis and inhibiting B-cell differentiation and proliferation [149]. Additionally, some members of the let-7 and miR-9 families with decreased levels in WM patients, in comparison with normal individuals, downregulate *PRDM1*, a significant regulator of B-cell development. Other miRNAs with increased levels in this malignancy such as miR-125b-5p and miR-181a-5p also target *PRDM1* and other factors contributing to B-cell development, including *IRF4* [149]. Let-7a-5p with lower levels in WM, compared to normal individuals, acts as an onco-suppressor by regulating different oncogenes such as *MYC* [142]. Conversely, miR-21-5p with increased levels in WM acts as an oncogene as it regulates numerous tumor suppressors, including *PTEN* and *PDCD4* [150]. In PCFCL, miR-9-5p, miR-129-2-3p, and miR-155-5p with upregulated levels in comparison to primary cutaneous DLBCL-leg type and cutaneous MZL are involved in normal B-cell development by targeting *PRDM1*, *SOX4*, and *SPI1*, respectively [148]. SOX4 is involved in the transition step from pro- to pre-B cell, whereas SPI1 and PRDM1 are transcription factors acting at later stages of B-cell development and regulating plasma cell differentiation. Therefore, all the aforementioned information highlights miRNAs with deregulated levels in these malignancies, appearing to have also a significant regulatory effect in disease onset and progression.

Knowing the miRNA targets and the respective effect of their activity is important as it sheds light on a part of the regulatory network. However, it is of high importance to u the regulatory network of as many miRNAs or other small non-coding RNAs with specific mRNAs as possible, in order to identify unique and significant pathogenic effects. In this approach, an extensive “screening” would reveal specific interactions between miRNAs and mRNAs that could be assessed in order to diagnose, predict, and cure the disease. Nevertheless, it is the balanced outcome of all distinct regulatory steps that determine the final outcome at a cellular level.

**Table 4 biomedicines-09-00333-t004:** miRNAs as candidate biomarkers in rare indolent B-cell NHLs.

Disease	miRNAs	Sample Origin	Expression	Potential Biomarker	References
WM	miR-363-3p, miR-206-3p, miR-494-3p, miR-155-5p, miR-184-3p, miR-542-3p	Bone marrow CD19^+^ B cells	Upregulated in WM vs. normal CD19^+^ B cells	Diagnosis	[81]
	miR-9-3p		Downregulated in WM vs. normal CD19^+^ B cells		
	miR-193b-3p, miR-126-3p, miR-181a-5p, miR-125b-5p, miR-451a	Bone marrow or peripheral blood CD19^+^ and CD138^+^ B cells	Upregulated in WM vs. CLL	Differential diagnosis	[149]
	miR-92a-3p, miR-223-3p, miR-92b-3p, miR-363-3p		Upregulated in WM vs. MM		
	miR-9-3p, miR-193b-3p, miR-182-5p, miR-152-3p		Downregulated in WM vs. MM		
	miR-21-5p, miR-142-3p		Upregulated in WM vs. normal B-lineage cells	Diagnosis	
	miR-182-5p, miR-152-3p, miR-373-5p, miR-575-3p		Downregulated in WM vs. normal B-lineage cells		
	Combination of miR-320a-3p and miR-320b-3p	Serum	Downregulated in WM vs. normal blood donors; downregulated in WM vs. MGUS and MM	Diagnosis, differential diagnosis	[151]
	miR-151-5p, let-7a-5p		Downregulated in WM vs. normal blood donors; downregulated in WM vs. MGUS		
	miR-21-5p, miR-192-5p, miR-320b-3p	Exosomes	Increases with disease progression	Prediction of disease progression	[152]
	let-7d-5p		Decreases with disease progression		
HCL	miR-363-3p, miR-708-5p	Peripheral blood B cells	Downregulated in HCL vs. CLL	Differential diagnosis	[146]
	miR-221-3p, miR-222-3p, miR-22-3p, miR-24-3p, miR-27a-3p, let-7b-5p	Peripheral blood CD19^+^ B cells	Upregulated in HCL vs. other B-cell malignancies; upregulated in HCL vs. normal B-lineage cells	Diagnosis, differential diagnosis	[147]
PCFCL	miR-150-5p, miR-155-5p	FFPE tissues	Upregulated in PCFCL vs. cutaneous MZL	Differential diagnosis	[117]
	miR-129-2-3p, miR-214-3p, miR-31-5p, miR-9-5p		Upregulated in PCFCL vs. primary cutaneous DLBCL-leg type		[148]

Abbreviations: CLL, chronic lymphocytic leukemia; DLBCL, diffuse large B-cell lymphoma; FFPE, formalin-fixed, paraffin-embedded; HCL, hairy cell leukemia; MGUS, monoclonal gammopathy of undetermined significance; MM, multiple myeloma; MZL, marginal zone lymphoma; PCFCL, primary cutaneous follicle center lymphoma; WM, Waldenström’s macroglobulinemia.

## 6. Interplay between Cytokines and miRNAs in B-Cell Malignancies

Cytokines compose a broad category of small proteins, including chemokines, interferons, interleukins (IL), lymphokines, and tumor necrosis factors (TNFs), which are important in cell signaling. They are produced and secreted by a variety of cells including stromal cells, fibroblasts, and endothelial cells. In the immune system, they are produced by leukocytes and exert their function on other leukocytes or tissues that express cytokine receptors. Several cytokines act on B cells and play key roles in the development, survival, differentiation, and/or proliferation of B cells. Additionally, certain chemokines are implicated in B-cell function, namely in antibody production, while the chemokine signaling regulates adhesion and migration, and hence, it is vital for B-cell survival and development [153].

Considering the key role of cytokines in normal physiology, their deregulation can assist in the development of B-cell malignancies. An interesting example is provided by the CXCR4/CXCL12 axis in FL. More specifically, this axis is especially important, since it regulates normal B-cell recirculation between GC zones, the bone marrow, and peripheral blood [154]. CXCR4 is a G-protein coupled chemokine receptor, to which the chemokine CXCL12 binds. A frequent characteristic of FL cells is the high expression of CXCR4. These elevated CXCR4 levels could be attributed to and/or explain the increased activity of proteins such as HIF1A, VEGFA, and signaling pathways including the PI3K/AKT, NFkB, and NOTCH. For instance, FL is often characterized by high levels of the transcription factor HIF1A, whose target genes include CXCR4 and the angiogenesis regulator, VEGF. Moreover, activation of the CXCR4-CXCL12 axis, in turn, promotes signaling through the PI3K/AKT and MAPK pathways and affects surface levels of CD20 and BCR signaling, leading to a proliferative and antiapoptotic phenotype of FL cells. MYC is one of the targets of PI3K/AKT and MAPK pathways that is activated via the aforementioned axis, which in turn has multiple and key effects in miRNA expression in FL, as previously discussed [47].

miRNAs have been shown to respond to dynamic micro-environmental cues and to regulate multiple functions of B-cell populations, including their survival, development, and activation. Thus, miRNA functions contribute not only to immune homeostasis, but also to the control of immune tolerance. Among the most important proteins whose expression is targeted by miRNAs, are the cytokines. Cytokines act as both key upstream signals and major functional outputs, and therefore, can affect miRNA levels, as well [155]. Every cell procedure takes place in the context of a regulatory network rather than a regulatory axis and these networks alter under pathological states. For instance, miR-21-5p is a well-known oncomiR, which is highly expressed in several B-cell NHLs and has been associated with resistance to apoptosis. During plasma cell differentiation, miR-21-5p expression is downregulated by PRDM1, a key molecule in the terminal differentiation of B cells and a tumor suppressor in several lymphoid neoplasms. The expression of *PRDM1* is upregulated by the transcription factor STAT3, which has previously been activated by IL21. Although STAT3 also enhances *MIR21* expression, it is not able to counteract the repression of the latter by PRDM1 [156]. However, in multiple myeloma, a different phenomenon is observed. More specifically, IL6 activates STAT3, which in turn promotes *MIR21* expression. The high levels of miR-21-5p contribute to the high proliferative rate and anti-apoptotic phenotype of malignant cells [157,158].

Another interesting example of cytokine-induced miRNAs has been observed in DLBCL. More specifically, it was shown that miR-155-5p levels were increased by TNFA, even though the molecular background behind this interaction is not known. In turn, TNFA-induced miR-155-5p inhibits the expression of *INPP5D*, a suppressor of the PI3K/AKT signaling pathway. The elevated activity of PI3K/AKT pathway can lead to increased cell proliferation and growth in malignant cells [159]. However, the interplay between cytokines and miRNAs has not been thoroughly investigated in indolent B-cell NHLs. Considering the key role of both cytokines and miRNAs in the development and progression of these malignancies, the investigation of their interactions is critical. Moreover, it would shed light in the molecular base of these diseases assisting in the development of targeted and more efficient therapeutic approaches.

## 7. Limitations

As previously analyzed, miRNAs are characterized by a great regulatory potential in indolent B-cell NHLs. However, our knowledge regarding their function in these malignancies remains limited and derives from individual research studies, since the majority of studies focus on the expression profiling of miRNAs. Therefore, a massive functional analysis is critical and will assist in the unraveling of the role of these tiny regulators in indolent B-cell NHLs. Moreover, it would be helpful if the miRNAs with a validated role in normal B-cell development were investigated in the context of indolent B-cell NHLs. Based on the current literature, we suggest potential regulatory interactions via which miRNAs with deregulated expression patterns in indolent B-cell NHLs can exert their role in these malignancies (Table 5).

So far, the mouse and other animal models provide important insights into human B-cell development and disease. However, several studies report intrinsic differences in gene expression and gene regulation between the human system and mouse model and more prominently in the immune system [160]. An interesting example of such differences is observed in gene expression, early after T-cell activation, under the effect of IL2. More precisely, differential IL2 transcription kinetics can inhibit splicing in mouse models, but not in humans [161]. Regarding B-cell development, it has been clearly stated that B-cell populations exist in different abundances between human and murine organisms, while they can have additional differences, including localization. Precisely, the identification of differences in the non-memory B-cell pools is important for understanding the differences in mechanisms that contribute to B-cell homeostasis in the two species and in translating information obtained from mouse models to studies of human disease [162]. However, the existing comparative studies of mouse and human B-cell development have focused on B-cell precursor populations and activated B cells [163]. Considering these differences between these organisms, the distinct expression pattern of miRNAs and regulatory networks can exist, as well. Therefore, a critical consideration is required when extrapolating mouse data to the human system in basic and translational research.

One of the major obstacles in the research of miRNAs is the fact that they act not in a regulatory axis, but as part of a complex regulatory network. More precisely, one miRNA is able to bind to multiple target genes, which subsequently affect several pathways, while simultaneously one target gene can be targeted by multiple miRNAs. Such examples are miR-150-5p and miR-155-5p, which have been shown to target multiple genes that regulate lymphomagenesis, creating an interaction network, while they have been correlated with several hematological malignancies, as well. Particularly, miR-155-5p has been characterized both as oncogenic and tumor-suppressive miRNA, depending on the cellular context, the intermolecular interactions, and the type of malignancy. Besides miR-155-5p, other miRNAs have also been characterized as double-edged swords complicating the miRNA functional research and pointing out the complexity of cell homeostasis. Due to this complex miRNA regulatory network, there are contradictory findings regarding the function of miRNAs in normal and pathological states; therefore, it is difficult for the researchers to reach a conclusion. The phenomenon of crosstalk between different signaling pathways further complicates the regulation of cellular processes in lymphocytes and consequently in lymphomas. B-cell lymphomas are often characterized by elevated molecular and phenotypical heterogeneity, even among the malignant cells comprising the tumor. This high heterogeneity could also be reflected in differential expression patterns of miRNAs even among cells of the same tumor and provides another potential explanation regarding the contradictory results concerning the role of miRNAs in malignant conditions [164,165].

Moreover, as aforementioned, miRNAs are ideal biomarker candidates; however, research in this field is still in its infancy, especially due to the lack of an efficient and cost-effective method for the accurate detection of miRNAs. One of the reasons why this has not yet been achieved is that features such as detection limits, range of concentrations in bodily fluids, and modulation depending on various parameters (age, gender, health/disease) have not clearly been established, yet. Additionally, the findings generally lack reproducibility. There are several discordances reported between different teams that have analyzed the same malignancy types. In order to resolve this issue, standardized protocols must be developed both for the initial stages of the process, like sample collection, transport, and storage, as well as data analysis for the diversity of technological methods used. Particularly for sample collection, it is critical that the sample size is large enough so that the result can be characterized as statistically valid [9,166]. Finally, in the majority of studies, it was not explicitly stated whether miRNAs were 5p or 3p, and their sequence was not provided. This creates ambiguity in future research, necessitating the implementation of the current nomenclature system in all future studies.

## 8. Future Perspectives

Although the expression profiles of miRNAs have been greatly investigated, further research is necessary to unravel the complex functional networks. This endeavor shall aid in the utilization of miRNAs as therapeutic targets (Figure 5). miRNAs can either promote tumor cell proliferation and hence act as oncogenic miRNAs, or suppress uncontrolled cell division, acting as tumor suppressors. According to these distinct properties, two main therapeutic strategies involving miRNAs have been developed. The first one introduces single-stranded antisense oligonucleotides, known as antimiRs or antagomiRs, that target an oncogenic miRNA, into the cell. The aforementioned interaction prevents the miRNA from binding to its target mRNA, resulting in unaffected protein expression levels. The second approach provides an artificial double-stranded RNA molecule, known as miRNA mimic, that imitates the naturally occurring pre-miRNA. This strategy attempts to restore the reduced innate expression levels of a tumor-suppressive miRNA. A major obstacle hindering translation into the clinic is the possible degradation of these agents by RNases. Therefore, chemically modified RNA nucleotides and molecules are being tested to increase stability and efficacy. Locked nucleic acid (LNA) nucleotides are most widely used, followed by the addition of 2′-O-methyl groups or phosphorothioate-like groups. Safe and efficient delivery inside the desirable cells, without triggering an immune response and by minimizing potential endosomal escape, is of equal importance, and a variety of liposomes and nanoparticles are being tested for this purpose. Besides stand-alone therapies, it would be interesting to investigate whether the combination of chemotherapy, radiotherapy, or immunotherapy with a miRNA-based therapy could be more beneficial for patients and/or overcome resistance to currently established therapeutic regimens [7,167,168,169].

A quite hot research topic is the reciprocal regulation between miRNAs and the epigenetic machinery. More specifically, miRNAs as a component of the epigenetic machinery are implicated in epigenetic regulation. At the same time, RNA and histone modifications and DNA methylation regulate miRNA expression, while epigenetic-related enzymes can be the target of miRNAs. All these findings have enlightened the researchers regarding the miRNA-epigenetic feedback loop. Several studies have associated the dysregulation of this miRNA-epigenetic feedback loop with the initiation and development of various diseases, including B-cell NHLs, and have demonstrated its potential for application in clinical diagnosis and prognosis. Particularly, the determination of the methylation profile of miRNA genes and the quantification of the expression of enzymes involved in epigenetic mechanisms affecting miRNA expression could constitute a powerful approach for diagnosis and prognosis, while drugs targeting epigenetic regulators have become a promising therapeutic strategy for several malignancies, including leukemia. Although this research field is still in its infancy, and further study is required for establishing miRNAs as pivotal modulators of epigenetic effects in clinical practice, it seems to be a promising research field with great potential [170].

Lastly, several challenges need to be overcome so the miRNAs are widely used as efficient biomarkers, as it has already been addressed in the limitations section. Furthermore, it would be very useful if some extracellular and/or circulating miRNAs found in bodily fluids were associated strongly with pathological states. The introduction of circulating miRNAs in clinical research as non-invasive biomarkers would be quite beneficial since non-invasive procedures are relatively convenient, fast, and not painful for the patients. Extracellular miRNAs can be stabilized via protein interactions, particularly AGO2, or via inclusion within extracellular vesicles, such as exosomes and microvesicles, and apoptotic bodies, that are thoroughly investigated for non-invasive biomarkers discovery. However, further research is required to uncover the exact secretion and stabilization mechanisms involved in each case, as well as the establishment of standardized detection and quantification protocols [171]. This is of major importance, because miRNAs show great promise in personalized medicine and could probably assist in patient stratification, selection of optimal treatment, and monitoring of therapeutic response for each individual [171].

## 9. Conclusions

Indolent B-cell NHLs compose a highly heterogeneous group of lymphomas with a high occurrence rate worldwide. Therefore, the elucidation of its molecular background and pathogenesis, in general, is considered quite critical. The recent advances have assisted towards this direction, but this research field is still in its infancy. miRNAs have been repeatedly investigated in the context of their usage as biomarkers or therapeutic targets of several malignancies, including indolent B-cell NHLs, due to their regulatory potential. These tiny regulators play a vital role in B-cell development and normal B-cell function, in general, and their deregulation could lead to fatal consequences for the cell. Among the most intriguing miRNAs are miR-150-5p, miR-155-5p, and those of the miR-17/92 cluster, as they are not only necessary for normal B-cell development, but are also implicated in the pathogenesis of the majority of the aforementioned malignancies. Several studies have attempted to analyze the expression levels of miRNAs in indolent B-cell NHLs, aiming to establish a miRNA signature, distinct for each malignant state. Even though progress has been achieved in this field, additional research is necessary in order to lead to more solid conclusions. Additionally, the regulatory networks via miRNAs function have to be unraveled. Despite all the aforementioned difficulties and limitations in miRNA research, their great regulatory potential is quite promising regarding the deciphering of cancer development and progression as well as the potential exploitation of miRNAs in therapy.

## Figures and Tables

**Figure 1 biomedicines-09-00333-f001:**
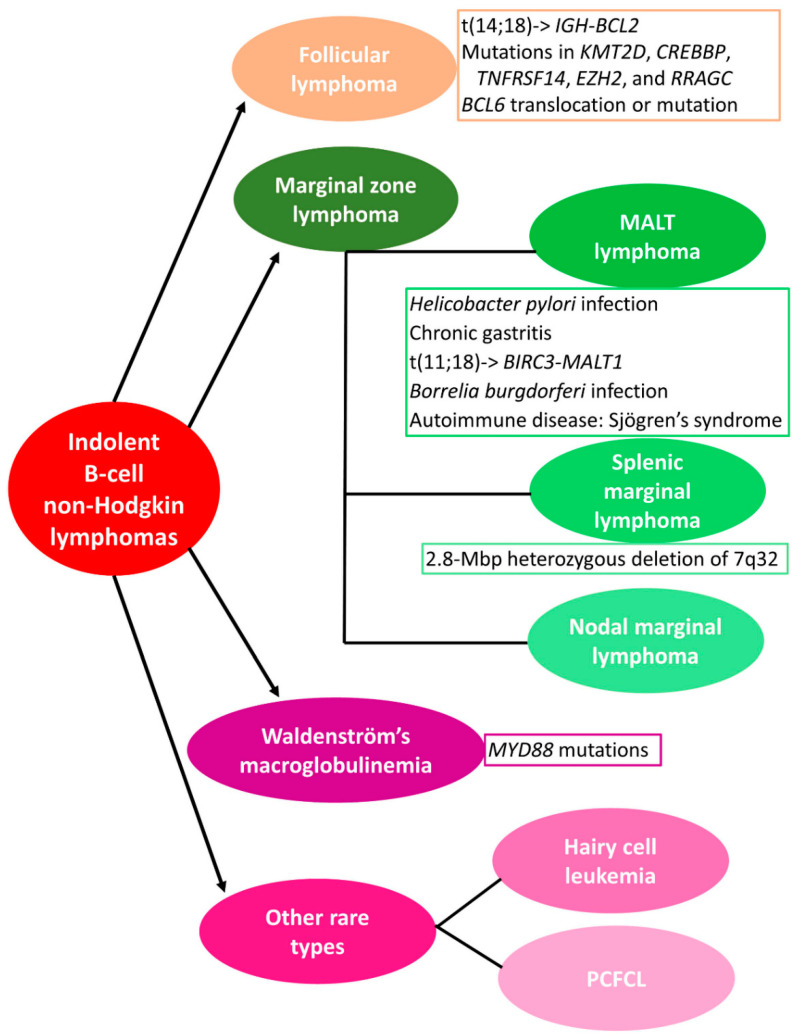
The main types of indolent B-cell non-Hodgkin lymphomas and some initiation factors of these malignancies. Abbreviations: MALT, mucosa-associated lymphoid tissue; PCFCL, primary cutaneous follicle center lymphoma.

**Figure 2 biomedicines-09-00333-f002:**
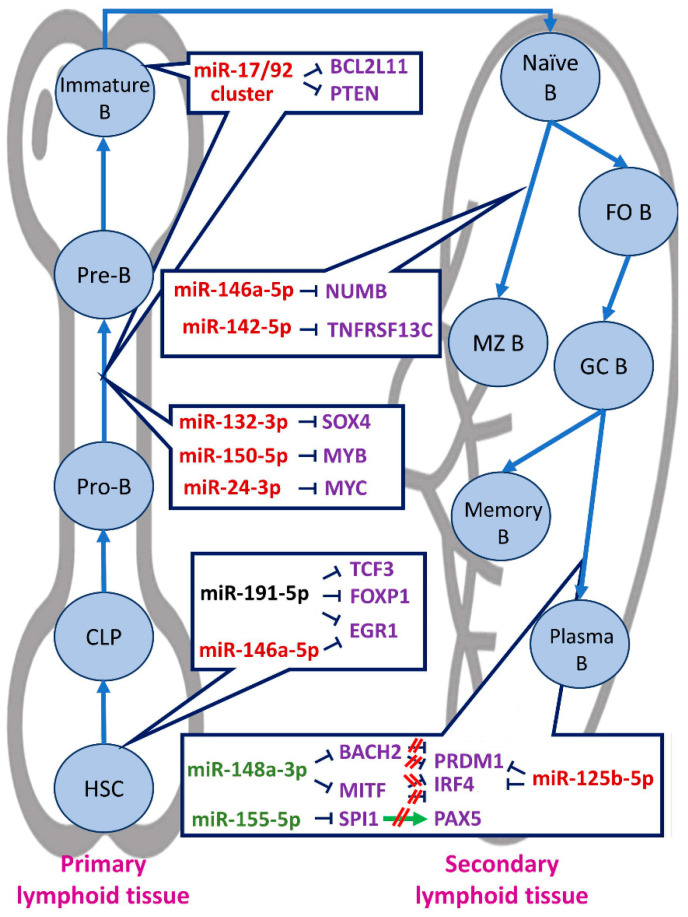
Brief illustration of B-cell development. miRNAs with a positive impact on the procedure are shown in green font, while those with a negative impact are shown in red. Black color indicates a miRNA with a controversial impact on B-cell development. miRNA targets are shown in purple font. Light blue arrows indicate the transition to the next developmental stage of B cells; dark blue “reverse tau” symbols (⊥) indicate attenuation of expression, whereas green arrows indicate induction of expression. Abbreviations: CLP, common lymphoid progenitor; FO B, follicular B cell; GC B, germinal center B cell; HSC, hematopoietic stem cell; MZ B, marginal zone B cell.

**Figure 3 biomedicines-09-00333-f003:**
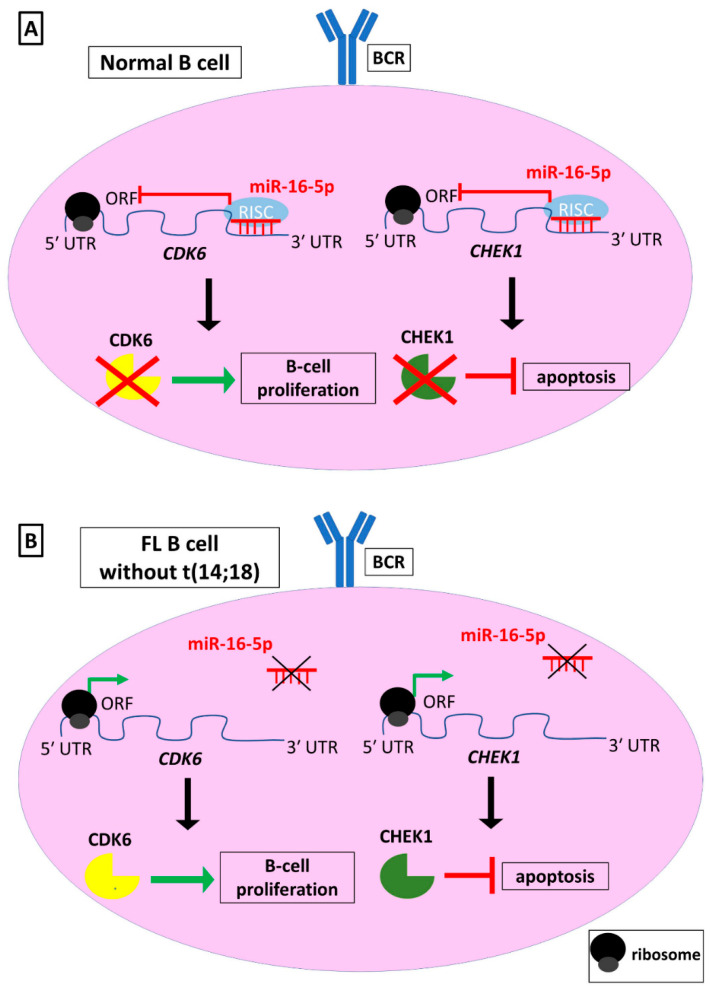
One of the potential effects of miR-16-5p in normal B cells (**A**) and follicular lymphoma (FL) B cells without t(14;18) (**B**). In the physiological state, its expression levels are high; miR-16-5p suppresses the expression levels of its target genes, *CHEK1* and *CDK6*, leading to apoptosis and cell cycle arrest. On the contrary, in a FL B cell, the expression levels of miR-16-5p are low, leading to increased expression of its targets. This results in B-cell apoptosis inhibition and increased proliferation rate, two hallmarks of a malignant cell. Black arrows indicate the transition to the next step; red “reverse tau” symbols (⊥) indicate inhibition, whereas green arrows indicate promotion of a cellular process. Abbreviations: BCR, B-cell receptor; RISC, RNA-induced silencing complex.

**Figure 4 biomedicines-09-00333-f004:**
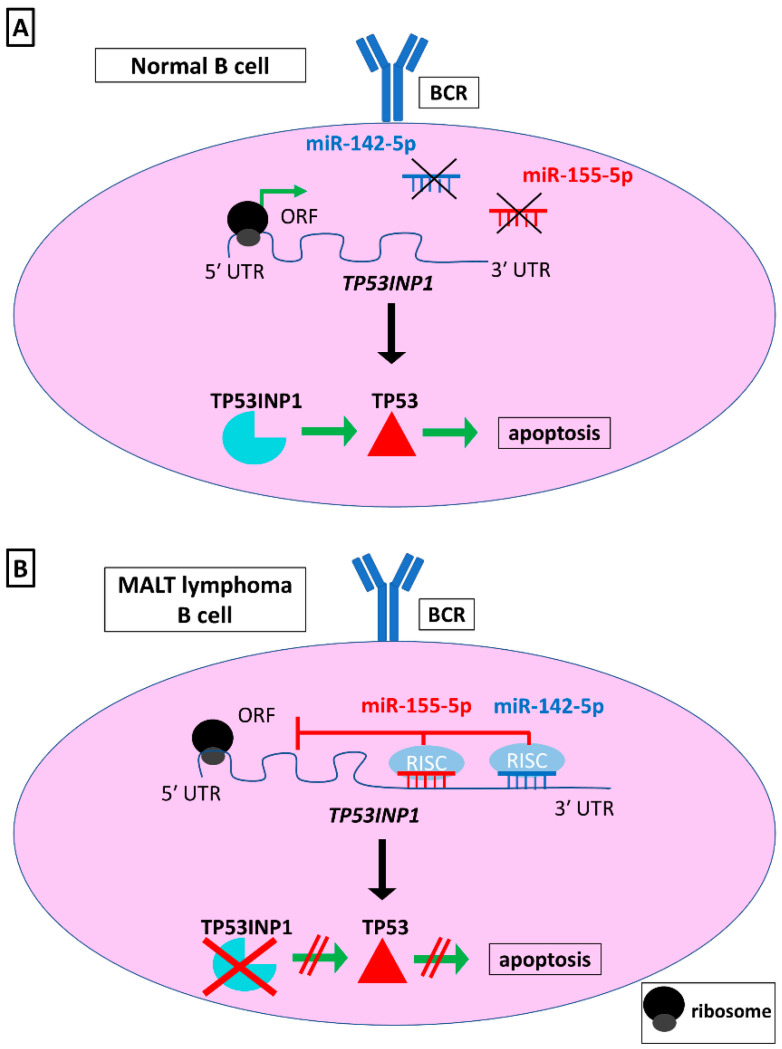
One of the potential effects of miR-155-5p and miR-142-5p in a normal B cell (**A**) and a mucosa-associated lymphoid tissue (MALT) lymphoma B cell (**B**). In physiological state, their expression levels are low and their target gene, *TP53INP1*, is expressed, leading to activation of TP53 and, subsequently, to apoptosis. On the contrary, in a MALT lymphoma B cell, the expression levels of miR-155-5p and miR-142-5p are increased, leading to decreased expression of their target. This results in apoptosis inhibition, one of the hallmarks of a malignant cell. Black arrows indicate transition to the next step; red “reverse tau” symbols (⊥) indicate inhibition, whereas green arrows indicate promotion of a cellular process. Abbreviations: BCR, B-cell receptor; RISC, RNA-induced silencing complex.

**Figure 5 biomedicines-09-00333-f005:**
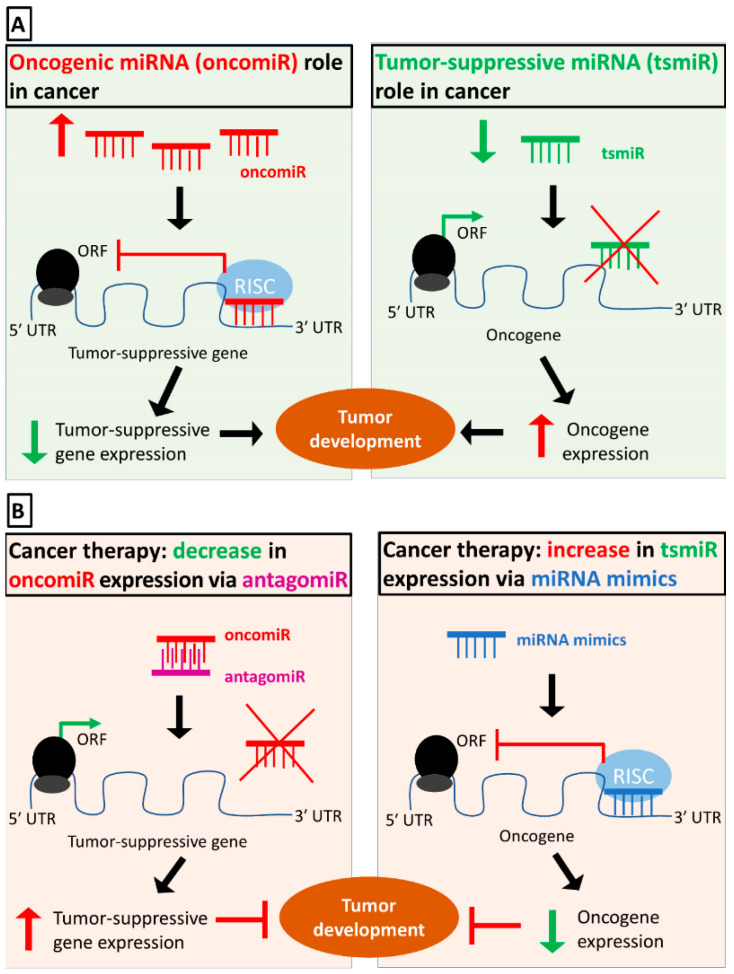
The role of oncogenic and tumor-suppressive miRNAs in cancer, and therapeutic strategies against cancer, based on miRNA targeting. (**A**) The oncogenic miRNAs are highly expressed in cancer; they bind to the 3′ UTR of their target genes (tumor suppressors), recruit RISC complex and suppress the expression of their target genes, leading to decreased levels of the respective proteins. In cancer, the tumor-suppressive miRNAs are expressed at low levels. Therefore, they are not able to suppress the expression levels of their target genes (oncogenes), leading to high levels of their proteins. (**B**) For the downregulation of oncogenic miRNAs and the subsequent attenuation of their harmful impact on cellular function, antagomiRs are used. They bind complementarily to the oncogenic miRNA and hence inhibit the binding of the latter to its target. For the upregulation of the tumor-suppressive miRNAs and the subsequent promotion of their beneficial impact on cell function, miRNA mimics are used. They have the same sequence as the specific tumor-suppressive miRNAs and, therefore, are able to bind to the targets of the latter and exert their function. Black arrows indicate the transition to the next step; red lines (**⊥)** indicate an inhibitory effect; upstream red arrows indicate an increase in the expression levels, while downstream green arrows indicate a decrease in the expression levels; horizontal green arrows indicate promotion of gene expression.

**Table 1 biomedicines-09-00333-t001:** miRNAs with a regulatory effect in indolent B-cell non-Hodgkin lymphomas (NHLs).

Disease	miRNAs	Expression in Lymphomas	Targets	Effect	References
Follicular lymphoma (FL)	miR-150-5p	Decreased	*FOXP1*	Inhibition of B-cell survival	[59]
	miR-31-5p		*E2F2*, *PIK3C2A*	Inhibition of cell cycle, survival, and migration	[62]
	miR-202-3p		*DICER1*	Regulation of biogenesis of miRNAs	[64]
			*SKP2*	Regulation of cell cycle transition	
	miR-618		*HDAC3*	Inhibition of cell cycle	[65]
			*CUL4A*	Inhibition of DNA damage response	
	miR-155-5p		*INPP5D*	Promotion of anti-tumor immune responses	[66]
	miR-16-5p		*CHEK1*	Promotion of B-cell apoptosis	[48]
			*CDK6*	Inhibition of B-cell proliferation	
	miR-20a-5p, miR-20b-5p	Increased	*CDKN1A*	Promotion of cell cycle	[63]
	miR-194-5p		*SOCS2*	Promotion of B-cell proliferation and survival	
	miR-93-5p		*MICA, MICB*	Inhibition of B-cell cytotoxicity	[67]
Gastric MALT lymphoma	miR-34a-5p	Decreased	*FOXP1*	Inhibition of B-cell survival	[22,68,69,70,71]
	miR-383-5p		*ZEB2*	Inhibition of epithelial-to-mesenchymal transition (EMT)	[72]
	miR-203a-3p		*ABL1*	Inhibition of B-cell proliferation	[73,74]
	miR-155-5p, miR-142-5p	Increased	*TP53INP1*	Inhibition of apoptosis	[74,75,76]
Splenic MZL	miR-26b-5p	Decreased	*NEK6*	Inhibition of mitosis-cell division	[77,78]
Waldenström’s macroglobulinemia (WM)	miR-9-3p	Decreased	*HDAC4, HDAC5*	Regulation of histone acetylation; Induction of WM cell cytotoxicity; promotion of WM cell autophagy and apoptosis	[79]
	miR-23b-3p		*SP1*	Suppression of NFkB signaling; Inhibition of cell proliferation and survival	[80]
	miR-155-5p	Increased	-	Promotion of MAPK/ERK, PI3K/AKT, and NFkB signaling; promotion of cell proliferation, adhesion, and migration	[81,82]
			*FOXO3, BCL2L11*	Inhibition of apoptosis	[83]
	miR-206-3p		*KAT6A*	Regulation of histone acetylation	[79]

Abbreviations: MALT, mucosa-associated lymphoid tissue; MZL, marginal zone lymphoma.

**Table 5 biomedicines-09-00333-t005:** Potential interactions between miRNAs that are deregulated in indolent B-cell NHLs and their validated targets in other malignancies.

Disease	miRNAs	Potential Target in Lymphomas	Potential Effect in B Cells	References
Gastric MALT lymphoma	miR-150-5p	*MYB*	Regulation of B-cell development	[14,76,100,101]
		*EGR2*	Inhibition of apoptosis	
	miR-196a-5p	*CDKN1B*	Promotion of cell cycle	[76,104]
	miR-153-3p	*AKT3*	Inhibition of cell proliferation	[76,105]
	miR-7-5p	*EGFR* and *IGF1R*	Inhibition of metastasis	[76,106,107]
	miR-16-5p	*BCL2*	Promotion of apoptosis	[113,114]
Splenic MZL	miR-29a/b1 cluster	*TCL1A*	Inhibition of cell proliferation	[126,128,129,130,131]
	miR-129-5p	*BCL2*	Promotion of apoptosis	
	miR-21-5p	*PTEN*, *FOXO3*	Promotion of cell proliferation, inhibition of apoptosis	[133,134]
Nodal MZL	miR-223-3p	*LMO2*	Inhibition of B-cell differentiation	[42,139]
OAL	miR-29 family	*TCL1A*	Inhibition of cell proliferation	[118,119]
		*CDK6*		
		*DNMT3B*	Inhibition of DNA methylation	
		*MCL1*	Promotion of apoptosis	
	miR-199a-5p	*IKBKB*	Promotion of apoptosis	[118,120,121]
WM	miR-9-3p	*PDRM1*	Inhibition of B-cell differentiation	[149]
	miR-125b-5p and miR-181a-5p	*PRDM1*, *IRF4*		
	let-7a-5p	*MYC*	Inhibition of cell proliferation	[142]
	miR-21-5p	*PTEN*, *PDCD4*	Promotion of cell proliferation, regulation of apoptosis	[150]
PCFCL	miR-9-5p	*PRDM1*	Inhibition of B-cell differentiation	[148]
	miR-129-2-3p	*SOX4*		
	miR-155-5p	*SPI1*		

Abbreviations: MALT, mucosa-associated lymphoid tissue; MZL, marginal zone lymphoma; OAL, Ocular adnexal lymphoma; PCFCL, primary cutaneous follicle center lymphoma; WM, Waldenström’s macroglobulinemia.

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
