# Peer review of "The Multifaceted Role and Utility of MicroRNAs in Indolent B-Cell Non-Hodgkin Lymphomas"

_biomedicines, 2021, doi:10.3390/biomedicines9040333_

Round 1

Reviewer 1 Report

The review by Artemaki et al. summarizes the multifunctional role of microRNAs (miRNAs) in indolent non-Hodgkin lymphoma (NHL). In general, the article is well written and interesting. The authors widely cite many key papers in the references and this would be helpful to update the information concerning the relationship between miRNAs and NHL; however, the story may be complicated, because each miRNA represses the expression of many target genes. Considering B-cell development, this manuscript mainly shed light on the biological functions of microRNAs in several indolent lymphomagenesis. As the authors state, follicular lymphoma (FL), one of indolent lymphomas, sometimes show malignant progression to diffuse large B-cell lymphoma (DLBCL) and the expression level of miR-31-5p decrease in DLBCL compared with FL. This point is clinically important to understand molecular basis of malignant transformation. Are there any cohort studies concerning miR-31-5p? As many studies have focused on expression profiling of miRNAs in several lymphomas, the biological significance still remains unsolved. Except for small number of knockout studies, it is hard to unveil in vivo function of miRNAs. For example, miR-155-5p has been considered as double-edged sword, oncogenic and tumor suppressive miRNA, depending on type of malignancies. In some cases, the information derived from cohort studies is presumably useful. Table 3 and 4 indicate that the expression of some miRNAs is associated with patients’ prognosis. If the data is presented in the cited papers, the authors should argue more precisely.

Author Response

Reviewer’s Comments and Corresponding Responses

Reviewer #1 (Comments to the Author):

 1.      The review by Artemaki et al. summarizes the multifunctional role of microRNAs (miRNAs) in indolent non-Hodgkin lymphoma (NHL). In general, the article is well written and interesting. The authors widely cite many key papers in the references and this would be helpful to update the information concerning the relationship between miRNAs and NHL; however, the story may be complicated, because each miRNA represses the expression of many target genes. Considering B-cell development, this manuscript mainly shed light on the biological functions of microRNAs in several indolent lymphomagenesis. As the authors state, follicular lymphoma (FL), one of indolent lymphomas, sometimes show malignant progression to diffuse large B-cell lymphoma (DLBCL) and the expression level of miR-31-5p decrease in DLBCL compared with FL. This point is clinically important to understand molecular basis of malignant transformation. Are there any cohort studies concerning miR-31-5p? As many studies have focused on expression profiling of miRNAs in several lymphomas, the biological significance still remains unsolved. Except for small number of knockout studies, it is hard to unveil in vivo function of miRNAs. For example, miR-155-5p has been considered as double-edged sword, oncogenic and tumor suppressive miRNA, depending on type of malignancies. In some cases, the information derived from cohort studies is presumably useful. Table 3 and 4 indicate that the expression of some miRNAs is associated with patients’ prognosis. If the data is presented in the cited papers, the authors should argue more precisely. We thank the Reviewer for these remarks. The regulatory networks of miRNAs are indeed complicated and necessitate further investigation.According to the Reviewer’s suggestion, we searched the current literature regarding potential functional role of the miRNAs presented in Tables 3 and 4. In the majority of the studies, only a single miRNA profiling was conducted, resulting in a potential miRNA signature. However, we recognize the need for connecting the miRNA deregulation with the deregulation of molecular changes in these malignancies. For this purpose, based on the existing literature regarding other malignancies, in the revised manuscript we suggest potential regulatory interactions via which miRNAs with deregulated expression patterns in indolent B-cell lymphomas can exert their role in these hematological malignancies. These results are also presented in Table 5.Additionally, we highlight the need for further investigation of the expression and the role miRNAs with key role in normal B-cell development in indolent B-cell non-Hodgkin lymphomas, as well. There are, also, some miRNAs which were deregulated in the majority of these diseases. This fact supports their potential key role in these malignancies. Therefore, we analyze the interactions of these specific miRNAs in indolent B-cell lymphomas more in depth, aiming to propose a potential regulatory axis. Besides miR-155-5p, we mention other miRNAs characterized as double-edge swords, as well. Page 14, Lines 458-465: On the contrary, miR-150-5p expression levels were low during FL transformation [59]. Considering these findings, it could be fruitful to investigate the functional role of miR-150-5p in MALT GL. Additionally, findings deriving from gastric cancer research revealed that miR-150-5p inhibits apoptosis in gastric cancer cells by targeting the pro-apoptotic gene EGR2 [101]. This molecule has been investigated in hematological malignancies and its deregulated expression or potential mutations have been associated with tumorigenesis [102,103]. Therefore, this could be a potential mechanism of action via which miR-150-5p could exert its role in MALT GL. Page 14, Lines 469-472: miR-150-5p and miR-155-5p have been associated with MALT GL in other studies, as has been aforementioned. The convergence of several research studies in the deregulated expression of these two miRNAs highlights their critical role in MALT GL. Page 15, Lines 496-509: During normal B-cell development, constitutive expression of miR-34a-5p can result in a block in B cell development at the pro-B cell to pre-B cell transition, leading to a re-duction in mature B cells. This block appeared to be mediated primarily by inhibited expression of the FOXP1 [110]. Several previous studies highlighted that FOXP1 is necessary for normal B-cell differentiation [22] and has been reported to predict MALT GL to gastric DLBCL transformation [70]. Indeed, data from another clinical study confirm that miR-34a-5p could be utilized as a prognostic biomarker to investigate MALT GL to gastric DLBCL transformation [71]. These results suggest a novel way that FOXP1 can lead to MALT GL progression, besides the t(3;14) chromosomal translocation that results in a FOXP1-IGH fusion gene and subsequently elevated levels of FOXP1 in MALT lymphomas [111,112]. Moreover, FOXP1 was elevated in high-grade transformation of FL [59]. All these findings highlight the pivotal role of FOXP1 in the development of B-cell malignancies and hence the role of miR-34a-5p as one of its potential regulators. Page 15, Lines 512-518: miR-16-5p is a key tumor-suppressive miRNA and has been repeatedly correlated with CLL and, as it has been aforementioned, with FL. Its more well-known target is the anti-apoptotic gene BCL2, via the suppression of which can inhibit apoptosis in CLL [114]. Even though its potential functional role in MALT GL transformation has not been unraveled, the existing data regarding its function in other hematological malignancies highlight its further investigation in this cancer, as well. Page 16, Lines 544-562: The majority of these miRNAs play a critical role in hematological malignancies and solid tumors, necessitating their further investigation in these malignancies, as well. Below, some potential modes of action of these miRNAs are proposed. One of the most critical miRNAs for further investigation is miR-16-5p, as it has been repeatedly characterized as a pivotal tumor suppressor in B-cell malignancies [113]. Furthermore, the members of the let-7 family can act as regulators of stem-cell differentiation and have also been implicated in tumor suppression in several ways. Interestingly, some members of this family suppress the acquisition and utilization of key nutrients, which are essential for B-cell activation. Additionally, members of the miR-29 family have been characterized as tumor suppressors in other malignancies, including mantle cell lymphoma, Burkitt lymphoma, and FL. This miRNA family is implicated in the regulation of several key pathways in carcinogenesis. Some of its main target genes are CDK6, DNMT3B, TCL1A, and MCL1, which are involved in cell cycle control, DNA methylation, and apoptosis inhibition, respectively [119]. Regarding miR-199a-5p, its high ex-pression has been associated with a better outcome in DLBCL patients, while one of its potential roles is the suppression of the NFkB signaling pathway, a critical pathway for the development of this malignancy [120,121]. Finally, miR-222-3p is another miRNA with a contradictory role since it has been characterized both as oncomiR and a tumor suppressor in lymphomas, highlighting the complex regulatory roles and networks of miRNAs. Page 16, Lines 572-576: The discovery of a potential implication of this miRNA in MALT lymphoma is quite important since until recently it was believed that it was degraded and the miR-200b-3p prevailed. However, recent data support the synergistic action of both miRNAs in the inhibition of EMT [125]. Therefore, the functional investigation of miR-200b-5p in MALT lymphoma and B-cell development in general could be interesting. Page 17, Lines 604-609: Even though the role of miR-21-5p has not been investigated in SMZL, there are sever-al studies that characterize it as oncomiR in NHLs, while there are studies, which ex-amine this miRNA as a therapeutic target. More precisely, it inhibits the expression of PTEN and FOXO3, molecules with a critical role in normal B-cell development, in hu-man B-NHL cell lines, while activates the PI3K/AKT pathway and provides resistance to chemotherapy in a human DLBCL cell line [134]. Page 17, Lines 618-625: NEK kinases, including NEK6, facilitate many mitotic events and, subsequently, cell division, while it is critical for STAT3 phosphorylation and hence, JAK/STAT signaling pathway [78]. Even though the overexpression of NEK6 has not been associated with SMZL, it has been associated with the development of other malignancies [138]. Thus, the decreased expression of miR-26b-5p and the subsequently increased expression of NEK6 in HCV-positive patients suggest a molecular mechanism of action through which HCV infection could lead to SMZL. Pages 17-18, Lines 640-648: FL and germinal center cells are distinguished by an increased expression of LMO2, and a diminished expression of miR-223-3p. In the aforementioned study, it was shown that the expression of LMO2 was low and the expression of miR-223-3p was high in NMZL patients, implicating a potential role of these molecules in NMZL development. Even though it is known that let-7f-5p is a member of the let-7 miRNA family, which has been shown to target various oncogenes and is usually underexpressed in many malignancies [142], a functional explanation of the differential ex-pression of this miRNA between these malignancies is not provided. Pages 20-21, Lines 749-791: Specific miRNAs with potential biomarker utility in the aforementioned rare types of indolent B-cell NHLs are summarized in Table 4. The identification of specific miRNA signatures with biomarker utility that can be used in order to distinguish specific rare types of indolent B-cell NHLs, from other types of leukemias, is of high im-portance for timely and optimum management of patients. Moreover, discovering miRNAs with biomarker utility may reveal other, promising molecules for further re-search. Even though all the aforementioned information is promising, further research is essential in order to elucidate the involvement of miRNAs in rare types of indolent B-cell NHLs. Elucidating the regulatory effect of miRNAs with different levels in rare indolent B-cell NHLs may reveal miRNAs-candidates that participate in pathogenic events which lead to these distinct malignancies. Characteristically, as previously mentioned, in all three types of rare indolent B-cell NHLs which are presented in this review, there are paradigms of miRNAs, having a potential biomarker utility, which can also have an oncogenic or an onco-suppressive role in each disease. Moreover, predicted targets of miR-363-3p, miR-494-3p, miR-184-3p and miR-542-3p, which are increased in WM patients include tumor suppressors, cell-cycle inhibitors, cytokine signaling suppressors, and tyrosine phosphatases [81]. miR-9-3p which acts as an onco-suppressor and is decreased in WM patients, targets protein kinases, oncogenes, and transcription factors enhancing apoptosis and inhibiting B-cell differentiation and cell proliferation [149]. Additionally, some members of let-7 and miR-9 families with de-creased levels in WM patients, in comparison with normal individuals downregulate PRDM1, a significant regulator of B-cell development. Other miRNAs such as miR-125b and miR-181a with increased levels in this malignancy also target PRDM1 and other factors contributing to B-cell development such as IRF4 [149]. Let-7a-5p with lower levels in WM, compared to normal individuals, acts as an onco-suppressor by regulating different oncogenes such as MYC [142]. Conversely, miR-21-5p with in-creased levels in WM, acts as an oncogene as it regulates numerous tumor suppressors, including PTEN and PDCD4 [150]. In PCFCL, miR-9-5p, miR-129-2-3p and miR-155-5p with upregulated levels in comparison to primary cutaneous DLBCL-leg type and cutaneous MZL are involved in normal B-cell development by targeting PRDM1, SOX4 and SPI1 respectively [148]. SOX4 is involved in the transition step from pro- to pre-B cell whereas SPI1 and PRDM1 are transcription factors which act in later steps of B-cell development regulating germinal center and plasma cell differentiation. Therefore, all the aforementioned information highlights miRNAs with deregulated levels in these malignancies which appear to have also a significant regulatory effect in disease onset and progression.Knowing the miRNA targets and the respective effect of their activity is important as it sheds light on a part of the regulatory network. However, it is of high im-portance to reveal the regulatory network of, as many as possible, miRNAs or other small non-coding RNAs with specific mRNAs in order to identify unique and significant pathogenic effects. In this approach, an extensive “screening” would reveal specific interactions between miRNAs and mRNAs that could be assessed in order to diagnose, predict, and cure the disease. Nevertheless, it is the balanced outcome of all distinct regulatory steps that determine the final outcome at a cellular level.”  Moreover, considering the Reviewer’s comment, we expanded the paragraph in which miR-31-5p is mentioned. More precisely, we have included additional cohort studies in which this miRNA was investigated, and we analyzed more deeply the molecular background of the transformation of FL to DLBCL, potentially orchestrated by miR-31-5p. Additionally, we analyzed in more detail the molecular basis of FL and the implication of miRNAs in this malignancy. Pages 8-9, Lines 294-313: Furthermore, it has been observed in a patient cohort study that as the disease progresses from FL to aggressive DLBCL, miR-31-5p expression levels decrease. This miRNA has attracted researchers’ interest due to its multifaceted role. It has been investigated in several malignant states and depending on its specific targets in distinct cell types, miR-31-5p can exert either an oncogenic or an onco-suppressive role. The low expression of this miRNA has been also observed in a cohort study with DLBCL patients [60]. This low expression can be achieved either by loss of the gene locus of this miRNA or by hypermethylation of its promoter, while both mechanisms have been detected in different malignancies. The MIR31 gene is located on chromosome band 9p21.3, 500 kb from the locus of the well-known tumor suppressors CDKN2A and CDKN2B. Because of their proximity, it is reasonable to suppose that MIR31 would be lost together with CDKN2A [61]. The deletion of the latter has been associated with poor prognosis of DLBCL patients. In a recent study regarding FL transformation, it was observed that E2F2 and PIK3C2A could be the direct targets of miR-31-5p. E2F2 is a transcription factor of the E2F family that permits the entry to S-phase, promoting cell cycle, and PIK3C2A is a catalytic subunit of the PI3K family, involved in cell migration, survival, and proliferation. High expression of E2F2 and elevated activity of PI3K/AKT pathway has been observed in DLBCL, while the latter has been associated with poor outcome of DLBCL patients, as well. Therefore, low miR-31-5p expression levels could result in a B-cell high-grade tumor, via the increased levels of the aforementioned proteins [62].” Page 5, Lines 189-211: FL is a broad and extremely complex clinical entity. Many genes and cellular pathways participate in the emergence and transformation of FL. In the majority of affected tissues, a t(14;18) translocation occurs, placing BCL2 locus next to the immunoglobulin heavy-chain enhancer and resulting in the constitutive expression of this anti-apoptotic protein [47]. However, FL development requires the acquisition of additional aberrations that enable proliferation, immune evasion, and support from microenvironmental factors. This is usually achieved by acquired aberrations in genes that control normal germinal center B-cell development.Precisely, in the early stages of development, FL cells acquire aberrations that enable them to (a) persist in germinal centers; (b) increase BCR signaling; (c) confer a “sustainable” level of genomic instability; and (d) inhibit apoptosis. These characteristics are achieved through the deregulation of a set of genes (KMT2D, CREBBP, TNFRSF14, EZH2, RRAGC), by acquiring mutations. However, these FL cells usually resemble centrocytes and similar to their normal counterparts, have a relatively low level of proliferation. The acquisition of aberrations that enable rapid proliferation, including MYC, the BCR pathway, the TLR pathway, the TP53 pathway, FOXO1, BCL6, alters the tumor nature, frequently leading to histologic transformation. Particularly, mutations and/or translocations in BCL6 locus are quite important in B-cell lympho-mas, since BCL6 is a transcription repressor, which targets a large range of genes, for instance, PRDM1, TP53, CDKN1A and BCL2, thus controlling the B-cell germinal formation, cell cycle and differentiation [48,49].So far, none of the current scoring systems and therapeutic approaches has been able to mitigate the risk of early progression or histologic transformation to DLBCL. Therefore, the discovery of novel biomarkers is of significant importance. Page 7, Lines 231-239: miR-16-5p has been associated with repression of the expression of BCL2 and hence, induction of apoptosis. Even though in t(14;18)–negative FL patients, miR-16-5p ex-pression is also associated with apoptosis, miR-16-5p exerts its role via an alternative regulatory network. More specifically, the decreased expression levels of this miRNA lead to increased expression of its target genes CHEK1, which encodes an apoptosis inhibitor and DNA repair monitor, and CDK6, which encodes a cyclin-dependent kinase and promoter of the cell cycle (Figure 3) [48]. These findings suggest a potential mechanism which could contribute to the pro-proliferative phenotype of t(14;18)–negative FLs.  Finally, we have included a paragraph in the limitations, in which we mention the differences with regard to gene expression regulation and B-cell development between the human organism and the mouse models, highlighting the carefulness needed when interpreting results from mouse models and expanding them to human organism. Page 23, Lines 864-880: So far, the mouse and other animal models provide important insights into human B-cell development and disease. However, several studies report intrinsic differences in gene expression and gene regulation between the human system and mouse model and more prominently in the immune system [160]. An interesting example of such differences is observed in gene expression, early after T-cell activation, under the effect of IL2. More precisely, differential IL2 transcription kinetics can inhibit splicing in mouse models but not in humans [161]. Regarding B-cell development, it has been clearly stated that B-cell populations exist in different abundance between human and murine organisms, while they can have additional differences, including localization. Precisely, the identification of differences in the non-memory B-cell pools is important for understanding the differences in mechanisms that contribute to B-cell homeostasis in the two species and in translating information obtained from mouse models to studies of human disease [162]. However, the existing comparative studies of mouse and human B-cell development have focused on B-cell precursor populations and activated B cells [163]. Considering these differences between these organisms, the dis-tinct expression pattern of miRNAs and regulatory networks can exist, as well. Therefore, a critical consideration is required when extrapolating mouse data to the human system in basic and translational research.  We have also included the relevant literature. 60.      Jardin, F.; Jais, J.P.; Molina, T.J.; Parmentier, F.; Picquenot, J.M.; Ruminy, P.; Tilly, H.; Bastard, C.; Salles, G.A.; Feugier, P., et al. Diffuse large B-cell lymphomas with CDKN2A deletion have a distinct gene expression signature and a poor prognosis under R-CHOP treatment: a GELA study. Blood 2010, 116, 1092-1104, doi:10.1182/blood-2009-10-247122.61.      Yu, T.; Ma, P.; Wu, D.; Shu, Y.; Gao, W. Functions and mechanisms of microRNA-31 in human cancers. Biomedicine & pharmacotherapy = Biomedecine & pharmacotherapie 2018, 108, 1162-1169, doi:10.1016/j.biopha.2018.09.132.86.      Battella, S.; Cox, M.C.; Santoni, A.; Palmieri, G. Natural killer (NK) cells and anti-tumor therapeutic mAb: unexplored interactions. Journal of leukocyte biology 2016, 99, 87-96, doi:10.1189/jlb.5VMR0415-141R.102.    Young, E.; Noerenberg, D.; Mansouri, L.; Ljungstrom, V.; Frick, M.; Sutton, L.A.; Blakemore, S.J.; Galan-Sousa, J.; Plevova, K.; Baliakas, P., et al. EGR2 mutations define a new clinically aggressive subgroup of chronic lymphocytic leukemia. Leukemia 2017, 31, 1547-1554, doi:10.1038/leu.2016.359.103.    Krysiak, K.; Gomez, F.; White, B.S.; Matlock, M.; Miller, C.A.; Trani, L.; Fronick, C.C.; Fulton, R.S.; Kreisel, F.; Cashen, A.F., et al. Recurrent somatic mutations affecting B-cell receptor signaling pathway genes in follicular lymphoma. Blood 2017, 129, 473-483, doi:10.1182/blood-2016-07-729954.110.    Rao, D.S.; O'Connell, R.M.; Chaudhuri, A.A.; Garcia-Flores, Y.; Geiger, T.L.; Baltimore, D. MicroRNA-34a perturbs B lymphocyte development by repressing the forkhead box transcription factor Foxp1. Immunity 2010, 33, 48-59, doi:10.1016/j.immuni.2010.06.013.119.    Kwon, J.J.; Factora, T.D.; Dey, S.; Kota, J. A Systematic Review of miR-29 in Cancer. Molecular therapy oncolytics 2019, 12, 173-194, doi:10.1016/j.omto.2018.12.011.120.    Wang, Q.; Ye, B.; Wang, P.; Yao, F.; Zhang, C.; Yu, G. Overview of microRNA-199a Regulation in Cancer. Cancer management and research 2019, 11, 10327-10335, doi:10.2147/CMAR.S231971.121.    Troppan, K.; Wenzl, K.; Pichler, M.; Pursche, B.; Schwarzenbacher, D.; Feichtinger, J.; Thallinger, G.G.; Beham-Schmid, C.; Neumeister, P.; Deutsch, A. miR-199a and miR-497 Are Associated with Better Overall Survival due to Increased Chemosensitivity in Diffuse Large B-Cell Lymphoma Patients. International journal of molecular sciences 2015, 16, 18077-18095, doi:10.3390/ijms160818077.125.    Rhodes, L.V.; Martin, E.C.; Segar, H.C.; Miller, D.F.; Buechlein, A.; Rusch, D.B.; Nephew, K.P.; Burow, M.E.; Collins-Burow, B.M. Dual regulation by microRNA-200b-3p and microRNA-200b-5p in the inhibition of epithelial-to-mesenchymal transition in triple-negative breast cancer. Oncotarget 2015, 6, 16638-16652, doi:10.18632/oncotarget.3184.134.    Fuertes, T.; Ramiro, A.R.; de Yebenes, V.G. miRNA-Based Therapies in B Cell Non-Hodgkin Lymphoma. Trends Immunol 2020, 41, 932-947, doi:10.1016/j.it.2020.08.006.138.    Jeon, Y.J.; Lee, K.Y.; Cho, Y.Y.; Pugliese, A.; Kim, H.G.; Jeong, C.H.; Bode, A.M.; Dong, Z. Role of NEK6 in tumor promoter-induced transformation in JB6 C141 mouse skin epidermal cells. The Journal of biological chemistry 2010, 285, 28126-28133, doi:10.1074/jbc.M110.137190.150.    Qi, L.; Bart, J.; Tan, L.P.; Platteel, I.; Sluis, T.; Huitema, S.; Harms, G.; Fu, L.; Hollema, H.; Berg, A. Expression of miR-21 and its targets (PTEN, PDCD4, TM1) in flat epithelial atypia of the breast in relation to ductal carcinoma in situ and invasive carcinoma. BMC cancer 2009, 9, 163, doi:10.1186/1471-2407-9-163.160.    Cheng, Y.; Ma, Z.; Kim, B.H.; Wu, W.; Cayting, P.; Boyle, A.P.; Sundaram, V.; Xing, X.; Dogan, N.; Li, J., et al. Principles of regulatory information conservation between mouse and human. Nature 2014, 515, 371-375, doi:10.1038/nature13985.161.    Bose, D.; Neumann, A.; Timmermann, B.; Meinke, S.; Heyd, F. Differential Interleukin-2 Transcription Kinetics Render Mouse but Not Human T Cells Vulnerable to Splicing Inhibition Early after Activation. Molecular and cellular biology 2019, 39, doi:10.1128/MCB.00035-19.162.    Heykers, A.; Leemans, A.; Van der Gucht, W.; De Schryver, M.; Cos, P.; Delputte, P. Differences in Susceptibility of Human and Mouse Macrophage Cell Lines to Respiratory Syncytial Virus Infection. Intervirology 2019, 62, 134-144, doi:10.1159/000502674.163.    LeBien, T.W.; Tedder, T.F. B lymphocytes: how they develop and function. Blood 2008, 112, 1570-1580, doi:10.1182/blood-2008-02-078071.  

The authors wish to thank the Reviewer for the constructive comments that led to the improvement of the manuscript.

Reviewer 2 Report

Artemaki et al. in this review article did a good job in describing the multifaceted roles of miRNAs in B-Cell non-Hodgkin Lymphomas. 

A key aspect of RNA biology that has been overlooked by the scientific community is the intrinsic differences in gene expression and gene regulation in the human system and mouse model (more prominently in the immune system). I would suggest the authors highlight this during the discussion or limitation part of the review article, with proper citations. The following is an example: https://pubmed.ncbi.nlm.nih.gov/31160491/  that can be cited here (along with other relevant references). 

Also, there are multiple cytokine/interferon-induced miRNA expressions that play a crucial role, it would be nice to have a discussion on that and their dysregulation (if any) in B-Cell non-Hodgkin Lymphomas.

Author Response

Reviewer’s Comments and Corresponding Responses

Reviewer #2 (Comments to the Author):

 1.      A key aspect of RNA biology that has been overlooked by the scientific community is the intrinsic differences in gene expression and gene regulation in the human system and mouse model (more prominently in the immune system). I would suggest the authors highlight this during the discussion or limitation part of the review article, with proper citations. The following is an example: https://pubmed.ncbi.nlm.nih.gov/31160491/ that can be cited here (along with other relevant references). Taking into consideration the Reviewer’s suggestion, we included a paragraph in the section of limitations in which we mention the phenomenon of differential gene expression and regulation pattern between the human organisms and mouse models.  Page 23, Lines 864-880: So far, the mouse and other animal models provide important insights into human B-cell development and disease. However, several studies report intrinsic differences in gene expression and gene regulation between the human system and mouse model and more prominently in the immune system [160]. An interesting example of such differences is observed in gene expression, early after T-cell activation, under the effect of IL2. More precisely, differential IL2 transcription kinetics can inhibit splicing in mouse models but not in humans [161]. Regarding B-cell development, it has been clearly stated that B-cell populations exist in different abundance between human and murine organisms, while they can have additional differences, including localization. Precisely, the identification of differences in the non-memory B-cell pools is important for understanding the differences in mechanisms that contribute to B-cell homeostasis in the two species and in translating information obtained from mouse models to studies of human disease [162]. However, the existing comparative studies of mouse and human B-cell development have focused on B-cell precursor populations and activated B cells [163]. Considering these differences between these organisms, the dis-tinct expression pattern of miRNAs and regulatory networks can exist, as well. There-fore, a critical consideration is required when extrapolating mouse data to the human system in basic and translational research. Additionally, we have added the relevant literature, including the one suggested by the Reviewer. 160.    Cheng, Y.; Ma, Z.; Kim, B.H.; Wu, W.; Cayting, P.; Boyle, A.P.; Sundaram, V.; Xing, X.; Dogan, N.; Li, J., et al. Principles of regulatory information conservation between mouse and human. Nature 2014, 515, 371-375, doi:10.1038/nature13985.161.    Bose, D.; Neumann, A.; Timmermann, B.; Meinke, S.; Heyd, F. Differential Interleukin-2 Transcription Kinetics Render Mouse but Not Human T Cells Vulnerable to Splicing Inhibition Early after Activation. Molecular and cellular biology 2019, 39, doi:10.1128/MCB.00035-19.162.    Heykers, A.; Leemans, A.; Van der Gucht, W.; De Schryver, M.; Cos, P.; Delputte, P. Differences in Susceptibility of Human and Mouse Macrophage Cell Lines to Respiratory Syncytial Virus Infection. Intervirology 2019, 62, 134-144, doi:10.1159/000502674.163.    LeBien, T.W.; Tedder, T.F. B lymphocytes: how they develop and function. Blood 2008, 112, 1570-1580, doi:10.1182/blood-2008-02-078071. 2.      Also, there are multiple cytokine/interferon-induced miRNA expressions that play a crucial role, it would be nice to have a discussion on that and their dysregulation (if any) in B-Cell non-Hodgkin Lymphomas. We wish to thank very much the Reviewer for this suggestion. We have searched the current literature regarding cytokine/interferon-induced miRNAs that play a crucial role in B-cell non-Hodgkin lymphomas, but this field is currently unexplored. However, we added in the revised manuscript a section in which we analyze the significance of cytokines in B-cell development and B-cell malignancies and the interplay between miRNAs and cytokines in hematological malignancies, related to indolent B-cell non-Hodgkin lymphomas.  Pages 21-22, Lines 797-852: 6. Interplay between cytokines and miRNAs in B-cell malignanciesCytokines are a broad category of small proteins, including chemokines, interferons, interleukins (IL), lymphokines, tumor necrosis factor (TNF), which are important in cell signaling. They are produced and secreted by a variety of cells including stromal cells, fibroblasts, and endothelial cells. In the immune system, they are produced by leukocytes and exert their function on other leukocytes or tissues that express the cytokine receptor. Several cytokines act on B cells and they play key roles in the development, survival, differentiation, and/or proliferation of B cells. Additionally, certain chemokines are implicated in B-cell function, namely in antibody production, while the chemokine signaling regulates adhesion and migration, and hence, it is vital for B-cell survival and development [153]. Considering the key role of cytokines in normal physiology, their deregulation can assist in the development of B-cell malignancies. An interesting example is provided by the CXCR4/CXCL12 axis in FL. More specifically, this axis is especially important, since it regulates normal B-cell recirculation between GC zones, the bone marrow, and peripheral blood [154]. CXCR4 is a G-protein coupled chemokine receptor, to which the chemokine CXCL12 binds. A frequent characteristic of FL cells is the high expression of CXCR4. These elevated CXCR4 levels could be attributed to and/or explain the in-creased activity of proteins such as HIF1A, VEGFA, and signaling pathways including the PI3K/AKT, NFkB, and NOTCH. For instance, FL is often characterized by high levels of the transcription factor HIF1A, whose target genes include CXCR4 and the angiogenesis regulator, VEGF. Moreover, activation of the CXCR4-CXCL12 axis, in turn, promotes signaling through the PI3K/AKT and MAPK pathways and affects surface levels of CD20 and BCR signaling, leading to a proliferative and antiapoptotic phenotype of FL cells. MYC is one of the targets of PI3K/AKT and MAPK pathways that is activated via the aforementioned axis, which in turn has multiple and key effects in miRNA expression in FL, as previously discussed [47].miRNAs have been shown to respond to dynamic micro-environmental cues and regulate multiple functions of B-cell populations including their development, survival, and activation. Thus, miRNA functions contribute not only to immune homeostasis but also to the control of immune tolerance. Among the most important proteins whose expression is targeted by miRNAs, are the cytokines. Cytokines act as both key upstream signals and major functional outputs, and therefore, can affect miRNA level, as well [155]. Every cell procedure takes place in the context of a regulatory network rather than a regulatory axis and these networks alter under pathological states. For instance, miR-21-5p is a well-known oncomiR, which is highly expressed in several B-cell NHLs and has been associated with resistance to apoptosis. During plasma cell differentiation, miR-21-5p expression is downregulated by PRDM1, a key molecule in the terminal differentiation of B cells and an oncosuppressor in several lymphoid neo-plasms. The expression of PRDM1 is upregulated by the transcription factor STAT3, which has been prior activated by IL21. Even though STAT3, also, promotes the MIR21 expression, it is not able to counteract the repression of the latter by PRDM1 [156]. However, in multiple myeloma, a different phenomenon is observed. More specifically, IL6 activates STAT3, which in turn promotes MIR21 expression. The high levels of miR-21-5p contribute to the high proliferative rate and anti-apoptotic phenotype of malignant cells [157,158]. Another interesting example of cytokine-induced miRNAs has been observed in DLBCL. More specifically, it was shown that miR-155-5p levels were increased by TNFA, even though the molecular background behind this interaction is not known. In turn, TNFA-induced miR-155-5p inhibits the expression of INPP5D, a suppressor of PI3K/AKT pathway. The elevated activity of PI3K/AKT pathway can lead to increased cell proliferation and growth in malignant cells [159]. However, the interplay between cytokines and miRNAs has not been thoroughly investigated in indolent B-cell NHLs. Considering the key role of both cytokines and miRNAs in the development and progression of these malignancies, the investigation of their interactions is critical. More-over, it would shed light in the molecular base of these diseases assisting in the development of targeted and more efficient therapeutic approaches.” Moreover, we have included the appropriate literature. 153.    Vazquez, M.I.; Catalan-Dibene, J.; Zlotnik, A. B cells responses and cytokine production are regulated by their immune microenvironment. Cytokine 2015, 74, 318-326, doi:10.1016/j.cyto.2015.02.007.154.    De Silva, N.S.; Klein, U. Dynamics of B cells in germinal centres. Nat Rev Immunol 2015, 15, 137-148, doi:10.1038/nri3804.155.    Garavelli, S.; De Rosa, V.; de Candia, P. The Multifaceted Interface Between Cytokines and microRNAs: An Ancient Mechanism to Regulate the Good and the Bad of Inflammation. Frontiers in immunology 2018, 9, 3012, doi:10.3389/fimmu.2018.03012.156.    Barnes, N.A.; Stephenson, S.; Cocco, M.; Tooze, R.M.; Doody, G.M. BLIMP-1 and STAT3 counterregulate microRNA-21 during plasma cell differentiation. Journal of immunology (Baltimore, Md. : 1950) 2012, 189, 253-260, doi:10.4049/jimmunol.1101563.157.    Loffler, D.; Brocke-Heidrich, K.; Pfeifer, G.; Stocsits, C.; Hackermuller, J.; Kretzschmar, A.K.; Burger, R.; Gramatzki, M.; Blumert, C.; Bauer, K., et al. Interleukin-6 dependent survival of multiple myeloma cells involves the Stat3-mediated induction of microRNA-21 through a highly conserved enhancer. Blood 2007, 110, 1330-1333, doi:10.1182/blood-2007-03-081133.158.    Papanota, A.M.; Karousi, P.; Kontos, C.K.; Ntanasis-Stathopoulos, I.; Scorilas, A.; Terpos, E. Multiple Myeloma Bone Disease: Implication of MicroRNAs in Its Molecular Background. International journal of molecular sciences 2021, 22, doi:10.3390/ijms22052375.159.    Pedersen, I.M.; Otero, D.; Kao, E.; Miletic, A.V.; Hother, C.; Ralfkiaer, E.; Rickert, R.C.; Gronbaek, K.; David, M. Onco-miR-155 targets SHIP1 to promote TNFalpha-dependent growth of B cell lymphomas. EMBO Mol Med 2009, 1, 288-295, doi:10.1002/emmm.200900028. Finally, we updated the literature in the revised manuscript by including additional key references: 60.      Jardin, F.; Jais, J.P.; Molina, T.J.; Parmentier, F.; Picquenot, J.M.; Ruminy, P.; Tilly, H.; Bastard, C.; Salles, G.A.; Feugier, P., et al. Diffuse large B-cell lymphomas with CDKN2A deletion have a distinct gene expression signature and a poor prognosis under R-CHOP treatment: a GELA study. Blood 2010, 116, 1092-1104, doi:10.1182/blood-2009-10-247122.61.      Yu, T.; Ma, P.; Wu, D.; Shu, Y.; Gao, W. Functions and mechanisms of microRNA-31 in human cancers. Biomedicine & pharmacotherapy = Biomedecine & pharmacotherapie 2018, 108, 1162-1169, doi:10.1016/j.biopha.2018.09.132.86.      Battella, S.; Cox, M.C.; Santoni, A.; Palmieri, G. Natural killer (NK) cells and anti-tumor therapeutic mAb: unexplored interactions. Journal of leukocyte biology 2016, 99, 87-96, doi:10.1189/jlb.5VMR0415-141R.102.    Young, E.; Noerenberg, D.; Mansouri, L.; Ljungstrom, V.; Frick, M.; Sutton, L.A.; Blakemore, S.J.; Galan-Sousa, J.; Plevova, K.; Baliakas, P., et al. EGR2 mutations define a new clinically aggressive subgroup of chronic lymphocytic leukemia. Leukemia 2017, 31, 1547-1554, doi:10.1038/leu.2016.359.103.    Krysiak, K.; Gomez, F.; White, B.S.; Matlock, M.; Miller, C.A.; Trani, L.; Fronick, C.C.; Fulton, R.S.; Kreisel, F.; Cashen, A.F., et al. Recurrent somatic mutations affecting B-cell receptor signaling pathway genes in follicular lymphoma. Blood 2017, 129, 473-483, doi:10.1182/blood-2016-07-729954.110.    Rao, D.S.; O'Connell, R.M.; Chaudhuri, A.A.; Garcia-Flores, Y.; Geiger, T.L.; Baltimore, D. MicroRNA-34a perturbs B lymphocyte development by repressing the forkhead box transcription factor Foxp1. Immunity 2010, 33, 48-59, doi:10.1016/j.immuni.2010.06.013.119.    Kwon, J.J.; Factora, T.D.; Dey, S.; Kota, J. A Systematic Review of miR-29 in Cancer. Molecular therapy oncolytics 2019, 12, 173-194, doi:10.1016/j.omto.2018.12.011.120.    Wang, Q.; Ye, B.; Wang, P.; Yao, F.; Zhang, C.; Yu, G. Overview of microRNA-199a Regulation in Cancer. Cancer management and research 2019, 11, 10327-10335, doi:10.2147/CMAR.S231971.121.    Troppan, K.; Wenzl, K.; Pichler, M.; Pursche, B.; Schwarzenbacher, D.; Feichtinger, J.; Thallinger, G.G.; Beham-Schmid, C.; Neumeister, P.; Deutsch, A. miR-199a and miR-497 Are Associated with Better Overall Survival due to Increased Chemosensitivity in Diffuse Large B-Cell Lymphoma Patients. International journal of molecular sciences 2015, 16, 18077-18095, doi:10.3390/ijms160818077.125.    Rhodes, L.V.; Martin, E.C.; Segar, H.C.; Miller, D.F.; Buechlein, A.; Rusch, D.B.; Nephew, K.P.; Burow, M.E.; Collins-Burow, B.M. Dual regulation by microRNA-200b-3p and microRNA-200b-5p in the inhibition of epithelial-to-mesenchymal transition in triple-negative breast cancer. Oncotarget 2015, 6, 16638-16652, doi:10.18632/oncotarget.3184.134.    Fuertes, T.; Ramiro, A.R.; de Yebenes, V.G. miRNA-Based Therapies in B Cell Non-Hodgkin Lymphoma. Trends Immunol 2020, 41, 932-947, doi:10.1016/j.it.2020.08.006.138.    Jeon, Y.J.; Lee, K.Y.; Cho, Y.Y.; Pugliese, A.; Kim, H.G.; Jeong, C.H.; Bode, A.M.; Dong, Z. Role of NEK6 in tumor promoter-induced transformation in JB6 C141 mouse skin epidermal cells. The Journal of biological chemistry 2010, 285, 28126-28133, doi:10.1074/jbc.M110.137190.150.    Qi, L.; Bart, J.; Tan, L.P.; Platteel, I.; Sluis, T.; Huitema, S.; Harms, G.; Fu, L.; Hollema, H.; Berg, A. Expression of miR-21 and its targets (PTEN, PDCD4, TM1) in flat epithelial atypia of the breast in relation to ductal carcinoma in situ and invasive carcinoma. BMC cancer 2009, 9, 163, doi:10.1186/1471-2407-9-163.

The authors wish to thank the Reviewer for the constructive comments that led to the improvement of the manuscript.

Round 2

Reviewer 2 Report

The authors did a good job in addressing the concerns and incorporating useful references. This has increased the quality of the manuscript and I think now it is ready for publication.